# BEST ARM IDENTIFICATION FOR STOCHASTIC RISING BANDITS

## ABSTRACT

Stochastic Rising Bandits (SRBs) model sequential decision-making problems in which the expected reward of the available options increases every time they are selected. This setting captures a wide range of scenarios in which the available options are *learning entities* whose performance improves (in expectation) over time (e.g., online best model selection). While previous works addressed the regret minimization problem, this paper focuses on the *fixed-budget Best Arm Identification* (BAI) problem for SRBs. In this scenario, given a fixed budget of rounds, we are asked to provide a recommendation about the best option at the end of the identification process. We propose two algorithms to tackle the above-mentioned setting, namely `R-UCBE`, which resorts to a UCB-like approach, and `R-SR`, which employs a successive reject procedure. Then, we prove that, with a sufficiently large budget, they provide guarantees on the probability of properly identifying the optimal option at the end of the learning process. Furthermore, we derive a lower bound on the error probability, matched by our `R-SR` (up to constant factors), and illustrate how the need for a sufficiently large budget is unavoidable in the SRB setting. Finally, we numerically validate the proposed algorithms in synthetic and realistic environments and compare them with the currently available BAI strategies.

## 1 INTRODUCTION

Multi-Armed Bandits (MAB, Lattimore & Szepesvári, 2020) are a well-known framework for sequential decision-making. Given a time horizon, the learner chooses, at each round, an option (i.e., arm) and observes a reward, which is a realization of an unknown distribution. The MAB problem is commonly studied in two flavors: *regret minimization* (Auer et al., 2002) and *best arm identification* (Bubeck et al., 2009). In regret minimization, the goal is to control the cumulative loss w.r.t. the optimal arm over a time horizon. Conversely, in best arm identification, the goal is to provide a recommendation about the best arm at the end of the time horizon. Specifically, we are interested in the fixed-budget scenario, where we seek to minimize the error probability of recommending the wrong arm at the end of the time budget, no matter the loss incurred during learning.

This work focuses on the *Stochastic Rising Bandits* (SRB), a specific instance of the *rested* bandit setting (Tekin & Liu, 2012) in which the expected reward of an arm increases whenever it is pulled. Online learning in such a scenario has been recently addressed from a regret minimization perspective by Metelli et al. (2022), providing no-regret algorithms for the SRB setting in both the rested and restless cases. The SRB setting models several real-world scenarios where arms improve their performance over time. A classic example is the so-called *Combined Algorithm Selection and Hyperparameter optimization* (CASH, Thornton et al., 2013; Kotthoff et al., 2017; Erickson et al., 2020; Li et al., 2020; Zöller & Huber, 2021), a problem of paramount importance in *Automated Machine Learning* (AutoML, Feurer et al., 2015; Yao et al., 2018; Hutter et al., 2019; Mussi et al., 2023). In CASH, the goal is to identify the *best learning algorithm* together with the *best hyperparameter* configuration for a given machine learning task (e.g., classification or regression). In this problem, every arm represents a hyperparameter tuner acting on a specific learning algorithm. A pull corresponds to a unit of time/computation in which we improve (on average) the hyperparameter configuration (via the tuner) for the corresponding learning algorithm (additional motivating examples in Appendix C). CASH was handled in a bandit *Best Arm Identification* (BAI) fashion in Li et al. (2020) and Cella et al. (2021). The former handles the problem by considering rising rested bandits

with *deterministic* rewards, failing to represent the intrinsic uncertain nature of such processes. Instead, the latter, while allowing stochastic rewards, assumes that the expected rewards evolve according to a *known* parametric functional class, whose parameters have to be learned.

**Original Contributions** In this paper, we address the design of algorithms to solve the BAI task in the rested SRB setting when a *fixed budget* is provided.[1] More specifically, we are interested in algorithms guaranteeing a sufficiently large probability of recommending the arm with the largest expected reward *at the end* of the time budget (as if only this arm were pulled from the beginning). The main contributions of the paper are summarized as follows:

- We propose two *algorithms* to solve the fixed-budget BAI problem in the SRB setting: `R-UCBE` (an optimistic approach, Section 4) and `R-SR` (a phases-based rejection algorithm, Section 5). First, we introduce specifically designed estimators required by the algorithms (Section 3). Then, we provide guarantees on the error probability of the misidentification of the best arm.
- We derive an error probability *lower bound* for the SRB setting, matched by our `R-SR` (up to constant factors), highlighting the complexity of the problem and the need for a sufficiently large time budget (Section 6).
- Finally, we conduct *numerical simulations* on synthetically generated data and a realistic online best model selection problem in comparison with state-of-art algorithms (Section 8).

The proofs of the statements are provided in Appendix E.

## 2 PROBLEM FORMULATION

**Stochastic Rising Bandits (SRBs)** We consider a rested Multi-Armed Bandit $\boldsymbol{\nu} = (\nu_i)_{i \in [\![K]\!]}$ with a finite number of arms $K$.[2] Let $T \in \mathbb{N}$ be the time budget; at every round $t \in [\![T]\!]$, the agent selects an arm $I_t \in [\![K]\!]$, plays it, and observes a reward $x_t \sim \nu_{I_t}(N_{I_t,t})$, where $\nu_{I_t}(N_{I_t,t})$ is the reward distribution of arm $I_t$ at round $t$ and depends on the number of pulls performed so far $N_{i,t} := \sum_{\tau=1}^{t} \mathbb{1}\{I_\tau = i\}$. The rewards are stochastic, formally $x_t := \mu_{I_t}(N_{I_t,t}) + \eta_t$, where $\mu_{I_t}(\cdot)$ is the expected reward of arm $I_t$ and $\eta_t$ is a zero-mean $\sigma^2$-subgaussian noise, conditioned to the past.[3] As customary, we assume that the expected rewards are bounded, formally $\mu_i(n) \in [0,1]$ for every $i \in [\![K]\!]$ and $n \in [\![T]\!]$. As in (Metelli et al., 2022), we focus on a particular family of rested bandits in which the expected rewards are monotonic *non-decreasing* and *concave* in expectation.

**Assumption 2.1** (Non-decreasing and concave expected rewards). *Let $\boldsymbol{\nu}$ be a rested MAB, defining $\gamma_i(n) := \mu_i(n+1) - \mu_i(n)$, for every $n \in [\![0, T]\!]$ and every arm $i \in [\![K]\!]$ the expected rewards are non-decreasing and concave, formally:*

$$\text{Non-decreasing:} \quad \gamma_i(n) \geqslant 0, \qquad\qquad \text{Concave:} \quad \gamma_i(n+1) \leqslant \gamma_i(n).$$

Intuitively, the $\gamma_i(n)$ represents the *increment* of the real process $\mu_i(\cdot)$ evaluated at the $n^{\text{th}}$ pull. Notice that concavity emerges in several settings, such as the best model selection and economics, representing the decreasing marginal returns (Lehmann et al., 2001; Heidari et al., 2016). A discussion on the learnability issues when the concavity assumption does not hold is provided in Appendix G.

**Learning Problem** The goal of fixed-budget BAI in the SRB setting is to select the arm providing the largest expected reward with a large enough probability given a fixed budget $T \in \mathbb{N}$. Unlike the stationary BAI problem (Audibert et al., 2010), in which the optimal arm is not changing, in this setting, we need to decide *when* to evaluate the optimality of an arm. We define optimality by considering the largest expected reward at time $T$. Formally, given a time budget $T$, the optimal arm $i^*(T) \in [\![K]\!]$, which we assume unique, satisfies $i^*(T) := \arg\max_{i \in [\![K]\!]} \mu_i(T)$, where we highlighted the dependence on $T$ as, with different values of the budget, $i^*(T)$ may change. Let $i \in [\![K]\!] \backslash \{i^*(T)\}$ be a suboptimal arm, we define the suboptimality gap as $\Delta_i(T) := \mu_{i^*(T)}(T) - \mu_i(T)$. We employ the notation $(i) \in [\![K]\!]$ to denote the $i^{\text{th}}$ best arm at time $T$ (breaking ties arbitrarily), i.e., $\Delta_{(2)}(T) \leqslant \cdots \leqslant \Delta_{(K)}(T)$. Given a rested MAB $\boldsymbol{\nu}$ and an algorithm $\mathfrak{A}$ that recommends $\hat{I}^*(T) \in [\![K]\!]$, we evaluate its performance with the *error probability*, i.e., the probability of recommending a suboptimal arm at the end of the time budget $T$: $e_T(\boldsymbol{\nu}, \mathfrak{A}) := \mathbb{P}_{\boldsymbol{\nu}, \mathfrak{A}}(\hat{I}^*(T) \neq i^*(T))$.

---

[1] We focus on the rested setting only and, thus, from now on, we will omit "rested" in the setting name.

[2] Let $y, z \in \mathbb{N}$, we denote with $[\![z]\!] := \{1, \ldots, z\}$, and with $[\![y, z]\!] := \{y, \ldots, z\}$.

[3] A zero-mean random variable $y \sim \nu$ is $\sigma^2$-subgaussian if it holds that $\mathbb{E}_{y \sim \nu}[e^{\xi y}] \leqslant e^{\frac{\sigma^2 \xi^2}{2}}$ for every $\xi \in \mathbb{R}$.

**Polynomial Increments** We now provide a characterization of a specific class of polynomial functions to upper bound the increments $\gamma_i(n)$.

**Assumption 2.2** (Polynomial $\gamma_i(n)$). *Let $\nu$ be a rested MAB, there exist $c > 0$ and $\beta > 1$ such that for every arm $i \in [\![K]\!]$ and number of pulls $n \in [\![0, T]\!]$ it holds that $\gamma_i(n) \leqslant cn^{-\beta}$.*

We anticipate that, even if our algorithms will not require this assumption, it will be used for deriving the lower bound and for providing more human-readable error probability guarantees.

## 3 ESTIMATORS

In this section, we introduce the estimators of the arm expected reward employed by the proposed algorithms. A visual representation of such estimators is provided in Figure 1.

Let $\varepsilon \in (0, 1/2)$, we employ an *adaptive arm-dependent window* size $h(N_{i,t-1}) := \lfloor \varepsilon N_{i,t-1} \rfloor$ to include the most recent samples only collected from arm $i \in [\![K]\!]$, avoiding the use of samples that are no longer representative. In the following, we design a *pessimistic* estimator and an *optimistic* estimator of the expected reward of each arm at the end of the budget time $T$, i.e., $\mu_i(T)$.[4]

**Pessimistic Estimator** The *pessimistic* estimator $\hat{\mu}_i(N_{i,t-1})$ is a negatively biased estimate of $\mu_i(T)$ obtained assuming that the function $\mu_i(\cdot)$ remains constant up to time $T$. This corresponds to the minimum admissible value under Assumption 2.1 (due to the *Non-decreasing* constraint). This estimator is an average of the last $h(N_{i,t-1})$ observed rewards collected from the $i^{\text{th}}$ arm, formally:

$$\hat{\mu}_i(N_{i,t-1}) := \frac{1}{h(N_{i,t-1})} \sum_{\tau=N_{i,t-1}-h(N_{i,t-1})+1}^{N_{i,t-1}} x_\tau. \tag{1}$$

The estimator enjoys the following concentration property.

**Lemma 3.1** (Concentration of $\hat{\mu}_i$). *Under Assumption 2.1, for every $a > 0$, simultaneously for every arm $i \in [\![K]\!]$ and number of pulls $n \in [\![0, T]\!]$, with probability at least $1 - 2TKe^{-a/2}$ it holds that:*

$$\hat{\beta}_i(n) - \hat{\zeta}_i(n) \leqslant \hat{\mu}_i(n) - \mu_i(n) \leqslant \hat{\beta}_i(n),$$

*where $\hat{\beta}_i(n) := \sigma\sqrt{\frac{a}{h(n)}}$ and $\hat{\zeta}_i(n) := \frac{1}{2}(2T - n + h(n) - 1)\gamma_i(n - h(n) + 1)$.*

As supported by intuition, we observe that the estimator is affected by a negative bias that is represented by $\hat{\zeta}_i(n)$ that vanishes as $n \to \infty$ under Assumption 2.1 with a rate that depends on the increment functions $\gamma_i(\cdot)$. Considering also the term $\hat{\beta}_i(n)$ and recalling that $h(n) = \mathcal{O}(n)$, under Assumption 2.2, the overall concentration rate is $\mathcal{O}(n^{-1/2} + cTn^{-\beta})$.

**Optimistic Estimator**[5] The *optimistic* estimator $\check{\mu}_i^T(N_{i,t-1})$ is a positively biased estimation of $\mu_i(T)$ obtained assuming that function $\mu_i(\cdot)$ linearly increases up to time $T$. This corresponds to the maximum value admissible under Assumption 2.1 (due to the *Concavity* constraint). The estimator is constructed by adding to the pessimistic estimator $\hat{\mu}_i(N_{i,t-1})$ an estimate of the increment occurring in the next steps up to $T$. The latter uses the last $2h(N_{i,t-1})$ samples to obtain an upper bound of such growth thanks to the concavity assumption, formally:

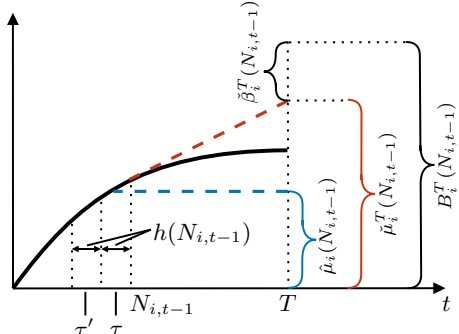

Figure 1: Graphical representation of the pessimistic $\hat{\mu}_i(N_{i,t-1})$ and the optimistic $\check{\mu}_i^T(N_{i,t-1})$ estimators.

$$\check{\mu}_i^T(N_{i,t-1}) := \hat{\mu}_i(N_{i,t-1}) + \sum_{\tau=N_{i,t-1}-h(N_{i,t-1})+1}^{N_{i,t-1}} (T-\tau)\frac{x_\tau - x_{\tau-h(N_{i,t-1})}}{h(N_{i,t-1})^2}. \tag{2}$$

The estimator displays the following concentration guarantee.

---

[4]Naïvely computing the estimators from their definition requires $\mathcal{O}(h(N_{i,t-1}))$ number of operations. An efficient way to incrementally update them, using $\mathcal{O}(1)$ operations, is provided in Appendix D.

[5]This estimator has appeared first in Metelli et al. (2022) and is here adapted for the BAI task.

**Lemma 3.2** (Concentration of $\check{\mu}_i^T$). *Under Assumption 2.1, for every $a > 0$, simultaneously for every arm $i \in [\![K]\!]$ and number of pulls $n \in [\![0, T]\!]$, with probability at least $1 - 2TKe^{-a/10}$ it holds that:*

$$\check{\beta}_i^T(n) \leqslant \check{\mu}_i^T(n) - \mu_i(n) \leqslant \check{\beta}_i^T(n) + \check{\zeta}_i^T(n),$$

*where $\check{\beta}_i^T(n) := \sigma \cdot (T - n + h(n) - 1)\sqrt{\frac{a}{h(n)^3}}$ and $\check{\zeta}_i^T(n) := \frac{1}{2}(2T - n + h(n) - 1)\gamma_i(n - 2h(n) + 1)$.*

Differently from the pessimistic estimation, the optimistic one displays a positive vanishing bias $\check{\zeta}_i^T(n)$. Under Assumption 2.2, we observe that the overall concentration rate is $\mathcal{O}(Tn^{-3/2} + cTn^{-\beta})$.

## 4 Optimistic Algorithm: Rising Upper Confidence Bound Exploration

We now introduce and analyze `Rising Upper Confidence Bound Exploration` (`R-UCBE`) an *optimistic* error probability minimization algorithm for the BAI task with a fixed budget for SRBs. The algorithm explores by means of a UCB-like approach and, for this reason, makes use of the optimistic estimator $\check{\mu}_i^T$ plus a bound to account for the uncertainty of the estimation. In `R-UCBE`, the choice of considering the optimistic estimator is natural and obliged since the pessimistic estimator is affected by negative bias and cannot be used to deliver optimistic estimates.

**Algorithm** `R-UCBE` (Algorithm 1) requires as input an exploration parameter $a \geqslant 0$, the window size $\varepsilon \in (0, 1/2)$, the time budget $T$, and the number of arms $K$. At first, it initializes to zero the counters $N_{i,0}$, and sets to $+\infty$ the upper bounds $B_i^T(N_{i,0})$ of all the arms (Line 2). Subsequently, at each round $t \in [\![T]\!]$, the algorithm selects the arm $I_t$ with the largest upper confidence bound (Line 4):

$$I_t \in \arg\max_{i \in [\![K]\!]} B_i^T(N_{i,t-1}) := \check{\mu}_i^T(N_{i,t-1}) + \check{\beta}_i^T(N_{i,t-1}), \tag{3}$$

$$\text{with:} \quad \check{\beta}_i^T(N_{i,t-1}) := \sigma \cdot (T - N_{i,t-1} + h(N_{i,t-1}) - 1)\sqrt{\frac{a}{h(N_{i,t-1})^3}}, \tag{4}$$

where $\check{\beta}_i^T(N_{i,t-1})$ represents the exploration bonus (a graphical representation is reported in Figure 1). Once the arm is chosen, the algorithm plays it and observes the feedback $x_t$ (Line 5). Then, the optimistic estimate $\check{\mu}_{I_t}^T(N_{I_t,t})$ and the exploration bonus $\check{\beta}_{I_t}^T(N_{I_t,t})$ of the selected arm $I_t$ are updated (Lines 8-9). The procedure is repeated until the algorithm reaches the time budget $T$. The final recommendation of the best arm is performed using the last computed values of the bounds $B_i^T(N_{i,T})$, returning the arm $\hat{I}^*(T)$ corresponding to the largest upper confidence bound (Line 12).

**Bound on the Error Probability of `R-UCBE`** We now provide bounds on the error probability for `R-UCBE`. We start with an analysis that considers no further characterization on the increments $\gamma_i(\cdot)$, beyond Assumption 2.1 and, then, we provide a more explicit result under Assumption 2.2.

**Theorem 4.1.** *Let $\nu$ be a rested MAB. Under Assumption 2.1, let $a^*$ be the largest positive value of $a$ satisfying:*

$$T - \sum_{i \neq i^*(T)} y_i(a) \geqslant 1, \tag{5}$$

*where for every $i \in [\![K]\!]$, $y_i(a)$ is the largest integer for which it holds:*

$$\underbrace{T\gamma_i(\lfloor(1 - 2\varepsilon)y\rfloor)}_{(A)} + \underbrace{2T\sigma\sqrt{\frac{a}{\lfloor\varepsilon y\rfloor^3}}}_{(B)} \geqslant \Delta_i(T). \tag{6}$$

*If $a^*$ exists, then for every $a \in [0, a^*]$ the error probability of `R-UCBE` is bounded by:*

$$e_T(\nu, \text{R-UCBE}) \leqslant 2TK\exp\left(-\frac{a}{10}\right). \tag{7}$$

Some comments are in order. First, $a^*$ is defined implicitly, depending on the constants $\sigma$, $T$, the increments $\gamma_i(\cdot)$, and the suboptimality gaps $\Delta_i(T)$. In principle, there might exist no $a^* > 0$ fulfilling condition in Equation 5 (this can happen, for instance, when the budget $T$ is not large enough), and, in such a case, we are unable to provide theoretical guarantees on the error probability of `R-UCBE`. Second, the result presented in Theorem 4.1 holds for generic increasing and concave expected reward functions (i.e., Assumption 2.1 only). This shows that, as expected, the error

**Algorithm 1:** R-UCBE.

**Input:** Time budget $T$, Number of arms $K$,
Window size $\varepsilon$, Exploration parameter $a$

1 Initialize $N_{i,0} = 0$,
2 $B_i^T(0) = +\infty, \forall i \in [\![K]\!]$
3 **for** $t \in [\![T]\!]$ **do**
4     Compute $I_t \in \arg\max_{i \in [\![K]\!]} B_i^T(N_{i,t-1})$
5     Pull arm $I_t$ and observe $x_t$
6     $N_{I_t,t} = N_{I_t,t-1} + 1$
7     $N_{i,t} = N_{i,t-1}, \quad \forall i \neq I_t$
8     Update $\check{\mu}_{I_t}^T(N_{I_t,t})$
9     Update $\check{\beta}_{I_t}^T(N_{I_t,t})$
10     Compute $B_{I_t}^T(N_{I_t,t}) = \check{\mu}_{I_t}^T(N_{I_t,t}) + \check{\beta}_{I_t}^T(N_{I_t,t})$
11 **end**
12 Recommend $\widehat{I}^*(T) \in \arg\max_{i \in [\![K]\!]} B_i^T(N_{i,T})$

**Algorithm 2:** R-SR.

**Input:** Time budget $T$, Number of arms $K$,
Window size $\varepsilon$

1 Initialize $t \leftarrow 1$, $N_0 = 0$, $\mathcal{X}_0 = [\![K]\!]$
2 **for** $j \in [\![K-1]\!]$ **do**
3     **for** $i \in \mathcal{X}_{j-1}$ **do**
4        **for** $l \in [\![N_{j-1} + 1, N_j]\!]$ **do**
5           Pull arm $i$ and observe $x_t$
6           $t \leftarrow t + 1$
7        **end**
8        Update $\hat{\mu}_i(N_j)$
9     **end**
10     Define $\overline{I}_j \in \arg\min_{i \in \mathcal{X}_{j-1}} \hat{\mu}_i(N_j)$
11     Update $\mathcal{X}_j = \mathcal{X}_{j-1} \backslash \{\overline{I}_j\}$
12 **end**
13 Recommend $\widehat{I}^*(T) \in \mathcal{X}_{K-1}$ (unique)

probability decreases when the exploration parameter $a$ increases until we reach the threshold $a^*$. Intuitively, the value of $a^*$ sets the maximum amount of exploration we should use for learning.

Under Assumption 2.2, i.e., using the polynomial characterization of the increment $\gamma_i(\cdot)$, we derive a result providing conditions on the time budget $T$ under which $a^*$ exists and its explicit value.

**Corollary 4.2.** *Let $\nu$ be a rested MAB. Under Assumptions 2.1 and 2.2, if the time budget $T$ satisfies:*

$$T \geqslant \begin{cases} \left( c^{\frac{1}{\beta}} (1-2\varepsilon)^{-1} \left( H_{1,1/\beta}(T) \right) + (K-1) \right)^{\frac{\beta}{\beta-1}} & \text{if } \beta \in (1, 3/2) \\ \left( c^{\frac{2}{3}} (1-2\varepsilon)^{-\frac{2}{3}\beta} \left( H_{1,2/3}(T) \right) + (K-1) \right)^3 & \text{if } \beta \in [3/2, +\infty) \end{cases}, \tag{8}$$

*there exists $a^* > 0$ defined as:*

$$a^* = \begin{cases} \frac{\varepsilon^3}{4\sigma^2} \left( \left( \frac{T^{1-1/\beta} - (K-1)}{H_{1,1/\beta}(T)} \right)^{\beta} - c(1-2\varepsilon)^{-\beta} \right)^2 & \text{if } \beta \in (1, 3/2) \\ \frac{\varepsilon^3}{4\sigma^2} \left( \left( \frac{T^{1/3} - (K-1)}{H_{1,2/3}(T)} \right)^{3/2} - c(1-2\varepsilon)^{-\beta} \right)^2 & \text{if } \beta \in [3/2, +\infty) \end{cases}, \tag{9}$$

*where $H_{1,\eta}(T) := \sum_{i \neq i^*(T)} \Delta_i(T)^{-\eta}$ for $\eta > 0$. Then, for every $a \in [0, a^*]$, the error probability of R-UCBE is bounded as in Equation 7.*

We notice that the error probability bound of Corollary 4.2 is the same as that of Theorem 4.1 and holds under the condition that the time budget $T$ fulfills Equation 8 (which is a recursive inequality in $T$). We defer a more detailed discussion on this condition to Remark 5.1, where we show that the existence of a finite value of $T$ fulfilling Equation 8 is ensured under mild conditions.

Let us remark that term $H_{1,\eta}(T)$ characterizes the complexity of the SRB setting, corresponding to term $H_1$ of Audibert et al. (2010) for the classical BAI problem when $\eta = 2$. As expected, in the small-$\beta$ regime (i.e., $\beta \in (1, 3/2)$), looking at the dependence of $H_{1,1/\beta}(T)$ on $\beta$, we realize that the complexity of a problem decreases as the parameter $\beta$ increases. Indeed, the larger $\beta$, the faster the expected reward reaches a stationary behavior. Nevertheless, even in the large-$\beta$ regime (i.e., $\beta \in [3/2, +\infty)$), the complexity of the problem is governed by $H_{1,2/3}(T)$, leading to an error probability larger than the corresponding one for BAI in standard bandits (Audibert et al., 2010). This can be explained by the fact that R-UCBE uses the optimistic estimator that, as shown in Section 3, enjoys a slower concentration rate compared to the standard sample mean, even for stationary bandits.

This two-regime behavior has an interesting interpretation when comparing Corollary 4.2 with Theorem 4.1. Indeed, $\beta = 3/2$ is the break-even threshold in which the two terms of the l.h.s. of Equation 6 have the same convergence rate. Specifically, the term $(A)$ takes into account the expected rewards growth (i.e., the bias in the estimators), while $(B)$ considers the uncertainty in the estimations of the R-UCBE algorithm (i.e., the variance). Intuitively, when the expected reward function displays a slow growth (i.e., $\gamma_i(n) \leqslant cn^{-\beta}$ with $\beta < 3/2$), the bias term $(A)$ dominates the variance term $(B)$ and the value of $a^*$ changes accordingly. Conversely, when the variance term $(B)$ is the dominant one (i.e., $\gamma_i(n) \leqslant cn^{-\beta}$ with $\beta > 3/2$), the threshold $a^*$ is governed by the estimation uncertainty,

being the bias negligible. Finally, we observe the trade-off introduced by the window size parameter $\varepsilon \in (0, 1/2)$. Indeed, by enlarging $\varepsilon$ we have a more demanding time budget requirement (Equation 8) but, at the same time, a smaller error probability since $a^*$ enlarges (Equation 9).

As common in optimistic algorithms for BAI (Audibert et al., 2010), setting a theoretically sound value of exploration parameter $a$ (i.e., computing $a^*$), requires knowledge of the setting, i.e., the complexity index $H_{1,\eta}(T)$.[6] In the next section, we propose an algorithm that relaxes this requirement.

## 5   PHASE-BASED ALGORITHM: RISING SUCCESSIVE REJECTS

In this section, we introduce the `Rising Successive Rejects` (R-SR), a phase-based solution inspired by the one proposed by Audibert et al. (2010), which overcomes the drawback of R-UCBE of requiring knowledge of $H_{1,\eta}(T)$.

**Algorithm** R-SR (Algorithm 2) takes as input the time budget $T$, the number of arms $K$, and the window size $\varepsilon$. At first, it initializes the set of the active arms $\mathcal{X}_0$ with all the available arms (Line 1), that will contain the arms that are still candidates to be recommended. The entire process proceeds through $K - 1$ phases. More specifically, during the $j^{\text{th}}$ phase, the arms still remaining in the active arms set $\mathcal{X}_{j-1}$ are played (Line 5) for $N_j - N_{j-1}$ times each, where:

$$N_j := \left\lceil \frac{1}{\overline{\log}(K)} \frac{T - K}{K + 1 - j} \right\rceil, \tag{10}$$

and $\overline{\log}(K) := \frac{1}{2} + \sum_{i=2}^{K} \frac{1}{i}$. At the end of each phase $j$, the arm with the smallest value of the pessimistic estimator $\hat{\mu}_i(N_j)$ is discarded from the set of active arms (Line 11). At the end of the $(K-1)^{\text{th}}$ phase, the algorithm recommends the (unique) arm remaining in $\mathcal{X}_{K-1}$ (Line 13).

It is worth noting that R-SR makes use of the pessimistic estimator $\hat{\mu}_i(n)$. Even if both estimators are viable for R-SR, the choice of the pessimistic estimator is justified by its better concentration rate $\mathcal{O}(n^{-1/2})$ compared to that of the optimistic one $\mathcal{O}(T n^{-3/2})$, being $n \leqslant T$ (see Section 3). Note that the phase lengths are the ones adopted by Audibert et al. (2010). This choice allows us to provide theoretical results without requiring domain knowledge (still under a large enough budget).

**Bound on the Error Probability of R-SR**   We now provide the error probability guarantees for R-SR. We start with a general analysis and, then, provide a more explicit result under Assumption 2.2.

**Theorem 5.1.** *Let $\boldsymbol{\nu}$ be a rested MAB. Under Assumption 2.1 if the time budget $T$ satisfies:*

$$\forall j \in [\![K-1]\!]: \qquad T\gamma_{(1)}(\lceil (1-\varepsilon)N_j \rceil) \leqslant \frac{\Delta_{(K-j+1)}(T)}{2}, \tag{11}$$

*then, the error probability of R-SR is bounded by:*

$$e_T(\boldsymbol{\nu}, R\text{-}SR) \leqslant \frac{K(K-1)}{2} \exp\left( -\frac{\varepsilon}{8\sigma^2} \cdot \frac{T - K}{\overline{\log}(K) H_2(T)} \right), \tag{12}$$

*where $H_2(T) := \max_{i \in [\![K]\!]} \left\{ i \Delta_{(i)}(T)^{-2} \right\}$ and $\overline{\log}(K) = \frac{1}{2} + \sum_{i=2}^{K} \frac{1}{i}$.*

The complexity of the problem is characterized by the term $H_2(T)$ that, for the standard MAB setting, reduces to the $H_2$ term of Audibert et al. (2010). Furthermore, when the condition of Equation 11 on the time budget $T$ is satisfied, the error probability coincides with that of the SR algorithm for standard MABs (apart for constant terms). We now provide a sufficient condition for the minimum time budget requirement of Equation 11 that holds under Assumption 2.2.

**Corollary 5.2.** *Let $\boldsymbol{\nu}$ be a rested MAB. Under Assumptions 2.1 and 2.2, if the time budget $T$ satisfies:*

$$T \geqslant \max\left\{ 2K, 2^{\frac{\beta+1}{\beta-1}} c^{\frac{1}{\beta-1}} (1-\varepsilon)^{-\frac{\beta}{\beta-1}} \overline{\log}(K)^{\frac{\beta}{\beta-1}} \max_{i \in [\![2,K]\!]} \left\{ i \Delta_{(i)}(T)^{-\frac{1}{\beta}} \right\}^{\frac{\beta}{\beta-1}} \right\}, \tag{13}$$

*then, the error probability of R-SR is bounded as in Equation 12.*

Similarly to the R-UCBE case, we observe a trade-off in the choice of $\varepsilon$ balancing the time budget requirement (Equation 13) and the error probability (Equation 12).

**Remark 5.1** (About the minimum time budget $T$). *To satisfy the $e_T$ bounds presented in Corollaries 4.2 and 5.2, R-UCBE and R-SR require the conditions of Equations 8 and 13 on the time budget*

---

[6]We defer the empirical study of the sensitivity of R-UCBE to $a$ to Section 8.

*T, respectively. If the suboptimal arms converge to an expected reward different from that of the optimal arm as $T \to +\infty$, it is always possible to find a finite value of $T < +\infty$ such that these conditions are fulfilled. Formally, assume that there exists $T_0 < +\infty$ and that for every $T \geqslant T_0$ we have that for all suboptimal arms $i \neq i^*(T)$ it holds that $\Delta_i(T) \geqslant \Delta_\infty > 0$. In such a case, the r.h.s. of Equations 8 and 13 are upper bounded by a function of $\Delta_\infty$ and are independent on $T$. Instead, if a suboptimal arm converges to the same expected reward as the optimal arm when $T \to +\infty$, the BAI problem is more challenging and, depending on the speed at which the two arms converge, the learning process slows down. This should not surprise as the BAI problem becomes non-learnable even in standard MABs with multiple optimal arms (Heide et al., 2021).*

## 6 LOWER BOUND

In this section, we investigate the complexity of the BAI problem for SRBs with a fixed budget.

**Minimum Time Budget $T$** We show that, under Assumptions 2.1 and 2.2, any algorithm requires a minimum time budget $T$ to be guaranteed to identify the optimal arm, even in a deterministic setting.

**Theorem 6.1.** *For every algorithm $\mathfrak{A}$, there exists a deterministic SRB $\boldsymbol{\nu}$ satisfying Assumptions 2.1 and 2.2 with $c = \beta - 1$ and $K \geqslant 1 + 8^{1/(\beta-1)}$ such that the optimal arm $i^*(T)$ cannot be identified (i.e., $e_T(\boldsymbol{\nu}, \mathfrak{A}) = 1$) for some time budgets $T$ unless:*

$$T \geqslant 8^{-\frac{1}{\beta-1}} H_{1,1/(\beta-1)}(T) = \sum_{i \neq i^*(T)} \left( \frac{1}{8\Delta_i(T)} \right)^{\frac{1}{\beta-1}}. \tag{14}$$

In the proof of Theorem 6.1, we show that each suboptimal arm $i \neq i^*(T)$ must be pulled at least an expected number of times of order $\mathbb{E}_{\boldsymbol{\nu}, \mathfrak{A}}[N_{i,T}] \geqslant \Omega\left(\Delta_i(T)^{-1/(\beta-1)}\right)$, formalizing the intuition that $i$ should be pulled sufficiently to ensure that, if pulled further, we are sure that it cannot become the optimal arm. It is worth comparing this bound on the time budget with the corresponding conditions on the minimum time budget requested by Equations 8 and 13 for R-UCBE and R-SR, respectively. Regarding R-UCBE, we notice that the minimum admissible time budget in the small-$\beta$ regime (i.e., $\beta \in (1, 3/2)$) is of order $H_{1,1/\beta}(T)^{\beta/(\beta-1)}$ which is larger than term $H_{1,1/(\beta-1)}(T)$ of Equation 14 (see Lemma E.12). Similarly, in the large-$\beta$ regime (i.e., $\beta \in [3/2, +\infty)$), the R-UCBE requirement is of order $H_{1,2/3}(T)^3 \geqslant H_{1,2}(T)$ which is larger than the term of Theorem 6.1 since $1/(\beta-1) < 2$.

Concerning R-SR, it is easy to show that $\max_{i \in [\![2,K]\!]} \{i\Delta_{(i)}(T)^{-1/\beta}\}^{\frac{\beta}{\beta-1}} \approx H_{1,1/\beta}(T)^{\frac{\beta}{\beta-1}}$, apart from logarithmic terms (see Lemma E.13), leading to a condition comparable to that of R-UCBE.[7]

**Error Probability Lower Bound** We now present a lower bound on the error probability that every algorithm performing fixed-budget BAI in the SRB setting suffers.

**Theorem 6.2.** *For every algorithm $\mathfrak{A}$ run with a time budget $T$ fulfilling Equation 14:*

- *there exists a set $\boldsymbol{\mathcal{V}}_1$ of SRBs satisfying Assumptions 2.1 and 2.2 with $K \geqslant 8^{1/(\beta-1)}$ and bounded complexity index $H_{1,2}(T) := \sum_{i \neq i^*(T)} \Delta_i(T)^{-2} \leqslant \overline{H} < +\infty$ such that:*

$$\sup_{\boldsymbol{\nu} \in \boldsymbol{\mathcal{V}}_1} e_T(\boldsymbol{\nu}, \mathfrak{A}) \geqslant \frac{1}{4} \exp\left( -\frac{8T}{\sigma^2 \overline{H}} \right);$$

- *there exists a set $\boldsymbol{\mathcal{V}}_2$ of SRBs satisfying Assumptions 2.1 and 2.2 with $K \geqslant 8^{1/(\beta-1)}$, complexity indexes $H_{1,2}(T) \leqslant \overline{H}$, and $\overline{H} \geqslant 16K^2$ such that:*

$$\sup_{\boldsymbol{\nu} \in \boldsymbol{\mathcal{V}}_2} e_T(\boldsymbol{\nu}, \mathfrak{A}) \exp\left( \frac{32T}{\sigma^2 \log(K) H_{1,2}(T)} \right) \geqslant \frac{1}{4}.$$

Some comments are in order. First, we stated the lower bound for the case in which the minimum time budget satisfies the inequality of Theorem 6.1, which is a necessary condition for identifying the optimal arm. Second, similarly to Carpentier & Locatelli (2016), we have that even when the algorithm is aware of a finite upper bound $\overline{H}$ on the complexity index $H_{1,2}(T)$, it will suffer an error probability at least of order $\Omega(\exp(-T/(\sigma^2 \overline{H})))$ in the worst-case problem. Furthermore, when we let the complexity index $H_{1,2}(T)$ take values depending on the number of arms $K$, the error

---

[7]For instance, when $\Delta_{(2)} = \ldots \Delta_{(K)} := \Delta$, the lower bound of Theorem 6.1 is of order $\Omega(K\Delta^{-1/(\beta-1)})$, while the requirements of R-UCBE and R-SR is of order $\tilde{\Omega}(K^{\beta/(\beta-1)}\Delta^{-1/(\beta-1)})$, showing a $K^{1/(\beta-1)}$ gap.

probability that any algorithm suffers is at least of order $\Omega(\exp(-T/(\sigma^2(\log K)H_{1,2}(T))))$. Thus, this second lower matches (up to constant factors) that of our R-SR since $H_2(T) \leqslant H_{1,2}(T)$ (see Lemma E.13), suggesting the superiority of this algorithm compared to R-UCBE. Finally, provided that the identificability condition of Equation 14, such a result corresponds to that of the standard (stationary) MABs (Audibert et al., 2010). A summary of the results is reported in Appendix A.

## 7 RELATED WORKS

As highlighted in Section 1, the works mostly related to ours are the ones by Li et al. (2020) and Cella et al. (2021) that both focus on the BAI problem in the rested setting given a fixed-budget (additional related works are discussed in Appendix B). Li et al. (2020) limits to *deterministic* arms and, thus, fails to capture the intrinsic stochasticity of the real-world processes they want to model. Furthermore, their analysis focuses on the *simple regret*: $r_T(\boldsymbol{\nu}, \mathfrak{A}) = \mu_{i*(T)}(T) - \mu_{\hat{I}*(T)}(N_{\hat{I}*(T),T})$, being $\hat{I}*(T) \in [\![K]\!]$ the recommendation of algorithm $\mathfrak{A}$, and the provided bound is expressed in terms of an implicit problem-dependent quantity that captures the rounds needed to distinguish the optimal arm from the others. Instead, Cella et al. (2021) deal with the problem of identifying the arm with the smallest loss which decreases as the arms are pulled. Clearly, such a setting can be transformed into an SRB by moving from losses to rewards. The authors consider the assumption that the expected loss follows a specific known parametric functional form: $\mu_i(n) = \frac{\alpha_i}{(1+N_{i,t-1})^\rho} + \beta_i$, where $\rho \in (0,1]$ is a *known* parameter and $\alpha_i$ and $\beta_i$ are to be estimated. This class of functions (with the flipped sign to consider rewards instead of losses) fulfills our Assumption 2.2 with $c = \rho\alpha_i$ and $\beta = 1 + \rho$. A lower bound on the expected simple regret of order $\Omega\left(T^{1-\beta}\right)$ is provided, that holds under certain conditions and for $K = 2$ (Theorem 1).[8] The Rest-Sure algorithm proposed in (Cella et al., 2021) requires the knowledge of $\rho$, which represents a major limitation since $\rho$ is usually unknown and hard to estimate in real-world scenarios. Instead, our R-SR works under Assumption 2.1 only with no further knowledge of the functional form of $\mu_i$.[9] Furthermore, Rest-Sure suffers a simple regret whose expression is quite convoluted (Theorem 4) but can be made explicit for some cases leading to the order $\widetilde{\mathcal{O}}\left(T^{1-\beta}\right)$. A simple analysis allows us to quantify the expected simple regret of our R-SR, as shown in Theorem E.10, that, under Assumptions 2.1 and 2.2, turns out to be of order $\widetilde{\mathcal{O}}\left(T^{1-\beta}\right)$, surprisingly matching the lower bound, up to logarithmic factors.

## 8 NUMERICAL VALIDATION

In this section, we provide a numerical validation of R-UCBE and R-SR. We compare them with state-of-the-art bandit baselines designed for stationary and non-stationary BAI in a synthetic setting, and we evaluate the sensitivity of R-UCBE to its exploration parameter $a$. Additional details about the experiments presented in this section are available in Appendix H. Additional experimental results on both synthetic settings and in a realistic experiment are available in Appendix I.[10]

**Baselines** We compare our algorithms against a wide range of solutions for BAI:

- RR: uniformly pulls all the arms until the budget ends in a *round-robin* fashion and, in the end, makes a recommendation based on the empirical mean of their reward over the collected samples;
- RR-SW: makes use of the same exploration strategy as RR to pull arms but makes a recommendation based on the empirical mean over the last $\frac{\varepsilon T}{K}$ collected samples from an arm.[11]
- UCB-E and SR (Audibert et al., 2010): algorithms for the stationary BAI problem;
- Prob-1 (Abbasi-Yadkori et al., 2018): an algorithm dealing with the adversarial BAI setting;
- ETC and Rest-Sure (Cella et al., 2021): algorithms developed for the decreasing loss BAI setting, that we converted through a linear transformation of the reward.

The hyperparameters required by the above methods have been set as prescribed in the original papers. For both our algorithms and RR-SW, we set $\varepsilon = 0.25$.

---

[8]When the suboptimality gap $\Delta$ used in the construction of Cella et al. (2021) is sufficiently small.

[9]Clearly, the error guarantees of R-SR will, then, depend on specific features of the functional of $\mu_i(\cdot)$, for instance, $c$ and $\beta$ when Assumption 2.2 holds, but the algorithm does not need to know them.

[10]The code to run the experiments is available in the supplementary material. It will be published in a public repository conditionally to the acceptance of the paper.

[11]The formal description of this baseline, as well as its theoretical analysis, is provided in Appendix F.

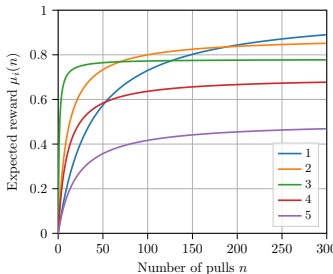
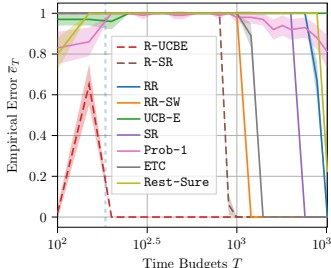
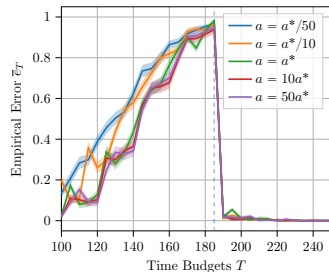

Figure 2: Expected values $\mu_i(n)$ for the arms of the synthetic setting.

Figure 3: Empirical error rate for the synthetically generated setting (100 runs, mean $\pm$ 95% c.i.).

Figure 4: Empirical error rate for the R-UCBE at different $a$ (1000 runs, mean $\pm$ 95% c.i.).

**Setting** To assess the quality of the recommendation $\hat{I}^*(T)$ provided by our algorithms, we consider a synthetic Gaussian SRB with $K = 5$ and $\sigma = 0.01$. Figure 2 shows the evolution of the expected rewards of the arms w.r.t. the number of pulls. In this setting, the optimal arm changes depending on whether $T \in [1, 185]$ or $T \in (185, +\infty)$. Thus, when the time budget is close to 185, the problem is more challenging since the optimal and second-best arms expected rewards are close to each other. For this reason, the BAI algorithms are less likely to provide a correct recommendation compared to when the time budgets make the expected rewards well separated. We compare the analyzed the algorithms $\mathfrak{A}$ in terms of empirical error $\bar{e}_T(\boldsymbol{\nu}, \mathfrak{A})$ (the smaller, the better), i.e., the empirical counterpart of $e_T(\boldsymbol{\nu}, \mathfrak{A})$ averaged over 100 runs, considering time budgets $T \in [100, 3200]$.

**Results** The empirical error probability provided by the analyzed algorithms in the synthetically generated setting is presented in Figure 3. We report with a dashed vertical blue line at $T = 185$, i.e., the budgets after which the optimal arm no longer changes. Before such a budget, all the algorithms provide large errors (i.e., $\bar{e}_T(\boldsymbol{\nu}, \mathfrak{A}) > 0.2$). However, R-UCBE outperforms the others by a large margin, suggesting that an optimistic estimator might be advantageous when the time budget is small. Shortly after $T = 185$, R-UCBE starts providing the correct suggestion consistently. R-SR begins to identify the optimal arm (i.e., with $\bar{e}_T(\boldsymbol{\nu}, \text{R-SR}) < 0.05$) for time budgets $T > 1000$. Nonetheless, both algorithms perform significantly better than the baseline algorithms used for comparison.

**Sensitivity Analysis for the Exploration Parameter of R-UCBE** We perform a sensitivity analysis on the exploration parameter $a$ of R-UCBE. Such a parameter should be set to a value less or equal to $a^*$, and the computation of the latter is challenging. We tested the sensitivity of R-UCBE to this hyperparameter by looking at the error probability for $a \in \{a^*/50, a^*/10, a^*, 10a^*, 50a^*\}$. Figure 4 shows the empirical errors of R-UCBE with different parameters $a$, where the blue dashed vertical line denotes the last time the optimal arm changes over the time budget. It is worth noting how, even in this case, we have two significantly different behaviors before and after such a time. Indeed, if $T \leq 185$, we have that misspecification with larger values than $a^*$ does not significantly impact the performance of R-UCBE, while smaller values slightly decrease the performance. Conversely, for $T > 185$ learning with different values of $a$ seems not to impact the algorithm performance significantly. This corroborates the previous results about the competitive performance of R-UCBE.

# 9 DISCUSSION AND CONCLUSIONS

This paper introduces the BAI problem with a fixed budget for the Stochastic Rising Bandits setting. Notably, such setting models many real-world scenarios in which the reward of the available options increases over time, and the interest is on the recommendation of the one having the largest expected rewards after the time budget has elapsed. In this setting, we presented two algorithms, namely R-UCBE and R-SR providing theoretical guarantees on the error probability. R-UCBE is an optimistic algorithm requiring an exploration parameter whose optimal value requires prior information on the setting. Conversely, R-SR is a phase-based solution that only requires the time budget to run. We established lower bounds for the error probability any algorithm suffers in such a setting, which is matched by our R-SR (up to constant factors). Furthermore, we showed how a requirement on the minimum time budget is unavoidable to ensure the identifiability of the optimal arm. Finally, we validate the performance of the two algorithms in both synthetically generated and real-world settings. A possible future line of research is to derive an algorithm balancing the trade-off between the error probability and the minimum requested time budget to properly identify the optimal arm.

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

## A  SUMMARY OF THE RESULTS

In this appendix, we summarize the theoretical results presented in this paper.

| | Time Budget $T$ |
|---|---|
| **LB** | $\sum_{i \neq i^*(T)} \left( \dfrac{1}{8\Delta_i(T)} \right)^{\frac{1}{\beta-1}}$ |
| **UB - R-UCBE** | $\begin{cases} \left( c^{\frac{1}{\beta}}(1-2\varepsilon)^{-1} \left( \sum_{i \neq i^*(T)} \dfrac{1}{\Delta_i^{1/\beta}(T)} \right) + (K-1) \right)^{\frac{\beta}{\beta-1}} & \text{if } \beta \in (1, 3/2) \\[2em] \left( c^{\frac{2}{3}}(1-2\varepsilon)^{-\frac{2}{3}\beta} \left( \sum_{i \neq i^*(T)} \dfrac{1}{\Delta_i^{2/3}(T)} \right) + (K-1) \right)^{3} & \text{if } \beta \in [3/2, +\infty) \end{cases}$ |
| **UB - R-SR** | $\max \left\{ 2K, 2^{\frac{\beta+1}{\beta-1}} c^{\frac{1}{\beta-1}}(1-\varepsilon)^{-\frac{\beta}{\beta-1}} \overline{\log}(K)^{\frac{\beta}{\beta-1}} \max_{i \in [\![2,K]\!]} \left\{ i\Delta_{(i)}(T)^{-\frac{1}{\beta}} \right\}^{\frac{\beta}{\beta-1}} \right\}$ |

Table 1: Bounds on the time budget. LB = lower bound, UB = upper bound.

| | Error Probability $e_T(\cdot)$ |
|---|---|
| **LB** | $\dfrac{1}{4} \exp\left( -\dfrac{32T}{\sigma^2 \log(K) \sum_{i \neq i^*(T)} \frac{1}{\Delta_i^2(T)}} \right)$ |
| **UB - R-UCBE** | $2TK \exp\left( -\dfrac{a}{10} \right)$ |
| **UB - R-SR** | $\dfrac{K(K-1)}{2} \exp\left( -\dfrac{\varepsilon}{8\sigma^2} \cdot \dfrac{T-K}{\overline{\log}(K) \max_{i \in [\![K]\!]} \left\{ i\Delta_{(i)}^{-2}(T) \right\}} \right)$ |

Table 2: Bounds on the error probability. LB = lower bound, UB = upper bound.

## B  ADDITIONAL RELATED WORKS

In this appendix, we integrate the related works presented in the main paper with the ones related to the *best arm identification* problem and the *rested bandit* setting.

**Best Arm Identification**  The pure exploration and BAI problems have been first introduced by Bubeck et al. (2009), while algorithms able to learn in such a setting have been provided by Audibert et al. (2010). The work by Gabillon et al. (2012) proposes a unified approach to deal with stochastic best arm identification problems by having either a fixed budget or fixed confidence. However, the stochastic algorithms developed in this line of research only provide theoretical guarantees in settings where the expected reward is stationary over the pulls. Abbasi-Yadkori et al. (2018) propose a method able to handle both the stochastic and adversarial cases, but they do not make explicit use of the properties (e.g., increasing nature) of the expected reward. Finally, (Garivier & Kaufmann, 2016; Kaufmann et al., 2016; Carpentier & Locatelli, 2016) analyze the problem of BAI from the lower bound perspective.

**Rested Bandits**  Bandit settings in which the evolution of an arm reward depends on the number of times the arm has been pulled, such as the one analyzed in our paper, are generally referred to as *rested*. A first general formulation of the rested bandit setting appeared in the work by Tekin & Liu (2012) and was further discussed by Mintz et al. (2020) and Seznec et al. (2020). In these works, the evolution of the expected reward of each arm is regulated by a Markovian process that is assumed to visit the same state multiple times. This is not the case for the rising bandits, where the

arm expected rewards continuously increase over the time budget. Finally, a specific instance of the rested bandits is constituted by the *rotting* bandits (Levine et al., 2017; Seznec et al., 2019; 2020), in which the expected payoff for a given arm decreases with the number of pulls. However, as pointed out by Metelli et al. (2022), techniques developed for this setting cannot be directly translated into ours, due to the inherently different nature of the problem.

## C  ADDITIONAL MOTIVATING EXAMPLES

In this appendix, we provide two additional motivating examples to better understand and appreciate the SRB setting.

**Selection of Athletes for Competitions**  Consider the role of a professional trainer for a team, having several athletes (i.e., our arms) to train in order to increase their performances. The final goal is to select a single athlete to represent the team in a competition. The performances of athletes increase when the trainer properly follows them. However, a trainer can follow just one athlete at a time. The trainer can be modeled as an agent performing best arm (athlete) identification, and the athletes represent the arms that increase their payoffs (i.e., performance) when pulled (i.e., when the trainer follows them).

**Online Best Model Selection**  Suppose we have to choose among a set of algorithms to maximize a given index (e.g., accuracy) over a training set. In this setting, we expect that all the algorithms progressively increase (on average) the index value and converge to their optimal value with different convergence rates. Therefore, we want to identify which candidate algorithm (i.e., arm) is the most likely to reach optimal performances, given the budget, and assign the available resources (e.g., computational power or samples). In summary, this problem reduces to the identification, with the largest probability, of the algorithm that converges faster to the optimum. A realistic example of such a scenario is provided in Figure 8.

## D  ESTIMATORS EFFICIENT UPDATE

In this appendix, we describe how to implement an efficient version (i.e., fully online) of the estimators we presented in the main paper. We resort to the update developed by Metelli et al. (2022). This update provides a way to achieve an $\mathcal{O}(1)$ computational complexity at each step for the update of the estimates for the pessimistic estimator $\hat{\mu}_i(t)$ and optimistic estimator $\check{\mu}_i^T(t)$.

With a slight abuse of notation, only in this appendix, for the sake of simplicity, we denote with $\overline{x}_{i,n}$ the reward collected at the $n^{\text{th}}$ pull from the arm $i$ and with $h_{i,t} = h(N_{i,t-1})$ the window size. Differently from the paper, here the reward subscript indicates the arm $i$ and the number of pulls of that arm $n$ instead of the total number of pulls $t$ we used in the definition of $x_t$.

More specifically, the *pessimistic* estimator $\hat{\mu}_i$ can be written as:

$$\hat{\mu}_i(t) = \frac{\overline{a}_i}{h_{i,t}},$$

where the quantity $\overline{a}_i$ is updated as follows:

$$\overline{a}_i \leftarrow \begin{cases} \overline{a}_i + \overline{x}_{i,N_{i,t}} - \overline{x}_{i,N_{i,t}-h_{i,t}} & \text{if } h_{i,t} = h_{i,t-1} \\ \overline{a}_i + \overline{x}_{i,N_{i,t}} & \text{otherwise} \end{cases},$$

and $\overline{a}_i = 0$ as the algorithm starts.

Instead, the *optimistic* estimator $\check{\mu}_i^T(t)$, is updated using:

$$\check{\mu}_i^T(t) = \frac{1}{h_{i,t}} \left( \overline{a}_i + \frac{T(\overline{a}_i - \overline{b}_i)}{h_{i,t}} - \frac{\overline{c}_i - \overline{d}_i}{h_{i,t}} \right).$$

Where the quantity $\bar{a}_i$ is defined and updated above and $\bar{b}_i$, $\bar{c}_i$, and $\bar{d}_i$ are updated as follows:

$$
\bar{b}_i \leftarrow \begin{cases} \bar{b}_i + \overline{x}_{i,N_{i,t}-h_{i,t}} - \overline{x}_{i,N_{i,t}-2h_{i,t}} & \text{if } h_{i,t} = h_{i,t-1} \\ \bar{b}_i + \overline{x}_{i,N_{i,t}-2h_{i,t}+1} & \text{otherwise} \end{cases},
$$

$$
\bar{c}_i \leftarrow \begin{cases} \bar{c}_i + N_{i,t} \cdot \overline{x}_{i,N_{i,t}} - (N_{i,t} - h_{i,t}) \cdot \overline{x}_{i,N_{i,t}-h_{i,t}} & \text{if } h_{i,t} = h_{i,t-1} \\ \bar{c}_i + N_{i,t} \cdot \overline{x}_{i,N_{i,t}} & \text{otherwise} \end{cases},
$$

$$
\bar{d}_i \leftarrow \begin{cases} \bar{d}_i + N_{i,t} \cdot \overline{x}_{i,N_{i,t}-h_{i,t}} - (N_{i,t} - h_{i,t}) \cdot \overline{x}_{i,N_{i,t}-2h_{i,t}} & \text{if } h_{i,t} = h_{i,t-1} \\ \bar{d}_i + (N_{i,t} - h_{i,t}) \cdot \overline{x}_{i,N_{i,t}-2h_{i,t}+1} + \bar{b}_i & \text{otherwise} \end{cases}.
$$

Similarly to what is presented above, the quantities are initialized to 0 as the algorithm start.

# E PROOFS AND DERIVATIONS

In this appendix, we provide all the proofs omitted in the main paper. For the sake of generality, we will provide the derivations for a generic choice of the window size of the estimators $h_{i,t}$ which depends on the arm $i \in [\![K]\!]$ and on the round $t \in [\![T]\!]$. When needed, we will particularize the choice for the case in which the window size depends on the number of pulls only, i.e., $h_{i,t} = h(N_{i,t-1})$.

## E.1 PROOFS OF SECTION 3

**Lemma E.1.** *Under Assumption 2.1, for every $i \in [\![K]\!]$, and $j, k \in \mathbb{N}$ with $k < j$, it holds that:*

$$
\gamma_i(j) \leqslant \frac{\mu_i(j) - \mu_i(k)}{j - k}.
$$

*Proof.* Using Assumption 2.1, we get:

$$
\gamma_i(j) = \frac{1}{j-k} \sum_{l=k}^{j-1} \gamma_i(j) \leqslant \frac{1}{j-k} \sum_{l=k}^{j-1} \gamma_i(l) = \frac{1}{j-k} \sum_{l=k}^{j-1} (\mu_i(l+1) - \mu_i(l)) = \frac{\mu_i(j) - \mu_i(k)}{j-k},
$$

where the first inequality comes from the concavity of the expected reward function (Assumption 2.1), and the second equality comes from the definition of increment. □

**Lemma E.2.** *For every arm $i \in [\![K]\!]$, every round $t \in [\![T]\!]$, and window size $1 \leqslant h_{i,t} \leqslant \lfloor N_{i,t-1}/2 \rfloor$, let us define:*

$$
\widetilde{\mu}_i^T(N_{i,t}) := \frac{1}{h_{i,t}} \sum_{l=N_{i,t-1}-h_{i,t}+1}^{N_{i,t-1}} \left( \mu_i(l) + (T-l)\frac{\mu_i(l) - \mu_i(l-h_{i,t})}{h_{i,t}} \right),
$$

*otherwise if $h_{i,t} = 0$, we set $\widetilde{\mu}_i^T(N_{i,t}) := +\infty$. Then, $\widetilde{\mu}_i^T(N_{i,t}) \geqslant \mu_i(T)$ and, if $N_{i,t-1} \geqslant 2$, it holds that:*

$$
\widetilde{\mu}_i^T(N_{i,t}) - \mu_i(T) \leqslant \frac{1}{2}(2T - 2N_{i,t-1} + h_{i,t} - 1)\,\gamma_i(N_{i,t-1} - 2h_{i,t} + 1).
$$

*Proof.* Following the derivation provided above, we have for every $l \in [\![2, \ldots, N_{i,T-1}]\!]$:

$$
\begin{aligned}
\mu_i(T) &= \mu_i(l) + \sum_{j=l}^{T-1} \gamma_i(j) \\
&\leqslant \mu_i(l) + (T-l)\,\gamma_i(l-1) \quad\quad\quad\quad\quad\quad (15) \\
&\leqslant \mu_i(l) + (T-l)\frac{\mu_i(l) - \mu_i(l-h_{i,t})}{h_{i,t}}, \quad\quad\quad (16)
\end{aligned}
$$

where Equation 15 follows from Assumption 2.1, and Equation 16 is obtained from Lemma E.1. By averaging over the most recent $1 \leqslant h_{i,t} \leqslant \lfloor N_{i,t-1}/2 \rfloor$ pulls, we get:

$$\mu_i(T) \leqslant \frac{1}{h_{i,t}} \sum_{l=N_{i,t-1}-h_{i,t}+1}^{N_{i,t-1}} \left( \mu_i(l) + (T-l)\frac{\mu_i(l)-\mu_i(l-h_{i,t})}{h_{i,t}} \right) =: \widetilde{\mu}_i^T(N_{i,t}).$$

For the bias bound, when $N_{i,t-1} \geqslant 2$, we retrieve:

$$\widetilde{\mu}_i^T(N_{i,t}) - \mu_i(T) = \frac{1}{h_{i,t}} \sum_{l=N_{i,t-1}-h_{i,t}+1}^{N_{i,t-1}} \left( \mu_i(l) + (T-l)\frac{\mu_i(l)-\mu_i(l-h_{i,t})}{h_{i,t}} \right) - \mu_i(T) \quad (17)$$

$$\leqslant \frac{1}{h_{i,t}} \sum_{l=N_{i,t-1}-h_{i,t}+1}^{N_{i,t-1}} (T-l)\frac{\mu_i(l)-\mu_i(l-h_{i,t})}{h_{i,t}}$$

$$= \frac{1}{h_{i,t}} \sum_{l=N_{i,t-1}-h_{i,t}+1}^{N_{i,t-1}} (T-l)\frac{1}{h_{i,t}} \sum_{j=l-h_{i,t}}^{l-1} \gamma_j(l)$$

$$\leqslant \frac{1}{h_{i,t}} \sum_{l=N_{i,t-1}-h_{i,t}+1}^{N_{i,t-1}} (T-l)\, \gamma_i(l-h_{i,t}) \quad (18)$$

$$\leqslant \frac{1}{2}(2T - 2N_{i,t-1} + h_{i,t} - 1)\, \gamma_i(N_{i,t-1} - 2h_{i,t} + 1), \quad (19)$$

where Equation 17 follows from Assumption 2.1 applied as $\mu_i(l) \leqslant \mu_i(N_{i,t})$, Equation 18 follows from Assumption 2.1 and bounding $\frac{1}{h_{i,t}} \sum_{j=l-h_{i,t}}^{l-1} \gamma_j(l) \leqslant \gamma_i(l-h_{i,t})$, and Equation 19 is still derived from Assumption 2.1 $\gamma_i(l-h_{i,t}) \leqslant \gamma_i(N_{i,t-1} - 2h_{i,t} + 1)$ and, finally, computing the summation. $\qquad\square$

**Lemma E.3.** *For every arm $i \in [\![K]\!]$, every round $t \in [\![T]\!]$, and window size $1 \leqslant h \leqslant \lfloor N_{i,t-1}/2 \rfloor$, let us define:*

$$\check{\mu}_i^T(N_{i,t}) := \frac{1}{h_{i,t}} \sum_{l=N_{i,t-1}-h_{i,t}+1}^{N_{i,t-1}} \left( X_i(l) + (T-l)\frac{X_i(l)-X_i(l-h_{i,t})}{h_{i,t}} \right),$$

$$\check{\beta}_i^T(N_{i,t}) = \sigma(T - N_{i,t-1} + h_{i,t} - 1)\sqrt{\frac{a}{h_{i,t}^3}},$$

*where $X_i(l)$ denotes the reward collected from arm $i$ when pulled for the $l$-th time. Otherwise, if $h_{i,t} = 0$, we set $\check{\mu}_i^T(t) := +\infty$ and $\check{\beta}_i^T(t) := +\infty$. Then, if the window size depends on the number of pulls only $h_{i,t} = h(N_{i,t-1})$, it holds that:*

$$\mathbb{P}\left( \exists t \in [\![T]\!] \, : \, \left| \check{\mu}_i^T(N_{i,t}) - \widetilde{\mu}_i^T(N_{i,t}) \right| > \check{\beta}_i^T(N_{i,t}) \right) \leqslant 2T \exp\left( -\frac{a}{10} \right).$$

*Proof.* Before starting the proof, it is worth noting that under the event $\{h_{i,t} = 0\}$, it holds that $\check{\mu}_i^T(t) = \widetilde{\mu}_i^T(t) = \check{\beta}_i^T(t) = +\infty$. Thus, under the convention that $\infty - \infty = 0$, then $0 > \check{\beta}_i^T(t)$ is not satisfied. For this reason, we need to perform our analysis under the event $\{h_{i,t} \geqslant 1\}$.

First, we remove the dependence on the number of pulls that, in a generic time instant, represents a random variable. Thus, we have:

$$\mathbb{P}\left( \exists t \in [\![T]\!] \, : \, \left| \check{\mu}_i^T(N_{i,t}) - \widetilde{\mu}_i^T(N_{i,t}) \right| > \check{\beta}_i^T(N_{i,t}) \right)$$

$$\leqslant \mathbb{P}\left( \exists n \in [\![0,T]\!] \text{ s.t. } h(n) \geqslant 1 \, : \, |\check{\mu}_i^T(n) - \widetilde{\mu}_i^T(n)| > \check{\beta}_i^T(n) \right)$$

$$\leqslant \sum_{n \in [\![0,T]\!]:h(n)\geqslant 1} \mathbb{P}\left( |\check{\mu}_i^T(n) - \widetilde{\mu}_i^T(n)| > \check{\beta}_i^T(n) \right), \quad (20)$$

where Equation 20 follows form a union bound over the possible values of $N_{i,t}$.

Now that we have a fixed value of $n$, consider a generic time $t$ in which arm $i$ has been pulled. We will observe a reward $x_t$ composed by the mean of the process $\mu_i(N_{i,t})$ plus some noise. The noise will be equal to $\eta_i(N_{i,t}) = x_t - \mu_i(N_{i,t})$, i.e., as the difference (not known) between the observed value for the arm $i$ at time $t$ and its real value at the same time. Let us rewrite the quantity to be bounded as follows for every $n$:

$$
h(n)\left(\check{\mu}_i^T(n) - \tilde{\mu}_i^T(n)\right) = \sum_{l=n-h(n)+1}^{n} \left(\eta_i(l) - (T-l)\cdot\frac{\eta_i(l) - \eta_i(l-h(n))}{h(n)}\right)
$$
$$
= \sum_{l=n-h(n)+1}^{n} \left(1 - \frac{T-l}{h(n)}\right)\cdot\eta_i(l) - \sum_{l=n-h(n)+1}^{n}\left(\frac{T-l}{h(n)}\right)\cdot\eta_i(l-h(n)).
$$

Here, notice that all the quantities $\eta_i(l)$ and $\eta_i(l-h(n))$ are independent since the number of pulls $l$ is fully determined by $n$ and $h(n)$, that now are non-random quantities.

Now, we apply the Azuma-Hoëffding's inequality of Lemma C.5 from Metelli et al. (2022) for weighted sums of subgaussian martingale difference sequences. To this purpose, we compute the sum of the square weights:

$$
\sum_{l=n-h(n)+1}^{n}\left(1 - \frac{T-l}{h(n)}\right)^2 + \sum_{l=n-h(n)+1}^{n}\left(\frac{T-l}{h(n)}\right)^2
$$
$$
\leq h(n)\cdot\left(1 + \frac{T-n+h(n)-1}{h(n)}\right)^2 + h(n)\cdot\left(\frac{T-n+h(n)-1}{h(n)}\right)^2
$$
$$
\leq \frac{5(T-n+h(n)-1)}{h(n)}.
$$

Given the previous argument, we have, for a fixed $n$:

$$
\mathbb{P}\left(|\check{\mu}_i^T(n) - \tilde{\mu}_i^T(n)| \geq \check{\beta}_i^T(n)\right)
$$
$$
\leq \mathbb{P}\left(\left|\sum_{l=n-h(n)+1}^{n}\left(1 - \frac{T-l}{h(n)}\right)\eta_i(l) - \sum_{l=n-h(n)+1}^{n}\left(\frac{T-l}{h(n)}\right)\eta_i(l-h(n))\right| \geq h(n)\check{\beta}_i^T(t)\right)
$$
$$
\leq 2\exp\left(-\frac{h(n)^2\check{\beta}_i^T(n)^2}{2\sigma^2\left(\frac{5(T-n+h(n)-1)}{h(n)}\right)}\right)
$$
$$
= 2\exp\left(-\frac{a}{10}\right).
$$

By replacing the obtained result into Equation 20 we get:

$$
\sum_{n\in[\![0,T]\!]:h(n)\geq 1} 2\cdot\exp\left(-\frac{a}{10}\right) \leq \sum_{n=1}^{t} 2\exp\left(-\frac{a}{10}\right) \leq 2T\exp\left(-\frac{a}{10}\right).
$$

$\square$

**Lemma E.4.** *For every arm $i \in [\![K]\!]$, every round $t \in [\![T]\!]$, and window size $1 \leq h_{i,t} \leq \lfloor N_{i,t-1}/2\rfloor$, let us define:*

$$
\bar{\mu}_i(N_{i,t}) := \frac{1}{h_{i,t}}\sum_{l=N_{i,t-1}-h_{i,t}+1}^{N_{i,t-1}}\mu_i(l),
$$

*otherwise, if $h_{i,t} = 0$, we set $\bar{\mu}_i(N_{i,t}) := 0$. Then, $\bar{\mu}_i(N_{i,t}) \leqslant \mu_i(T)$ and, if $N_{i,t-1} \geqslant 2$, it holds that:*

$$\mu_i(T) - \bar{\mu}_i(N_{i,t}) \leqslant \frac{1}{2}(2T - 2N_{i,t-1} + h_{i,t} - 1)\gamma_i(N_{i,t-1} - h_{i,t} + 1).$$

*Proof.* Following the derivation provided above, we have for every $l \in \{2, \ldots, N_{i,T-1}\}$:

$$\mu_i(T) = \mu_i(l) + \sum_{j=l}^{T-1} \gamma_i(j) \geqslant \mu_i(l). \tag{21}$$

Thus, by averaging over the most recent $1 \leqslant h_{i,t} \leqslant \lfloor N_{i,t-1}/2 \rfloor$ pulls, we get:

$$
\begin{aligned}
\mu_i(T) &= \frac{1}{h_{i,t}} \sum_{l=N_{i,t-1}-h_{i,t}+1}^{N_{i,t-1}} \left( \mu_i(l) + \sum_{j=l}^{T-1} \gamma_i(j) \right) \\
&= \bar{\mu}_i(N_{i,t}) + \frac{1}{h_{i,t}} \sum_{l=N_{i,t-1}-h_{i,t}+1}^{N_{i,t-1}} \sum_{j=l}^{T-1} \gamma_i(j) \\
&\leqslant \bar{\mu}_i(N_{i,t}) + \frac{1}{h_{i,t}} \sum_{l=N_{i,t-1}-h_{i,t}+1}^{N_{i,t-1}} \sum_{j=l}^{T-1} \gamma_i(j) \\
&\leqslant \bar{\mu}_i(N_{i,t}) + \frac{1}{2}(2T - 2N_{i,t-1} + h_{i,t} - 1)\gamma_i(N_{i,t-1} - h_{i,t} + 1),
\end{aligned}
$$

where we used Assumption 2.1. $\qquad\square$

**Lemma E.5.** *For every arm $i \in [\![K]\!]$, every round $t \in [\![T]\!]$, and window size $1 \leqslant h \leqslant \lfloor N_{i,t-1}/2 \rfloor$, let us define:*

$$\hat{\mu}_i^T(N_{i,t}) := \frac{1}{h_{i,t}} \sum_{l=N_{i,t-1}-h_{i,t}+1}^{N_{i,t-1}} X_i(l),$$

$$\hat{\beta}_i^T(N_{i,t}) = \sigma \sqrt{\frac{a}{h_{i,t}}},$$

*where $X_i(l)$ denotes the reward collected from arm $i$ when pulled for the $l$-th time. Otherwise, if $h_{i,t} = 0$, we set $\hat{\mu}_i^T(t) := +\infty$ and $\hat{\beta}_i^T(t) := +\infty$ . Then, if the window size depends on the number of pulls only $h_{i,t} = h(N_{i,t-1})$, it holds that:*

$$\mathbb{P}\left( \exists t \in [\![T]\!] \,:\, |\hat{\mu}_i(N_{i,t}) - \bar{\mu}_i(N_{i,t})| > \hat{\beta}_i(N_{i,t}) \right) \leqslant 2T \exp\left( -\frac{a}{2} \right).$$

*Proof.* Before starting the proof, it is worth noting that under the event $\{h_{i,t} = 0\}$, it holds that $\hat{\mu}_i^T(t) = \bar{\mu}_i^T(t) = \hat{\beta}_i^T(t) = +\infty$. Thus, under the convention that $\infty - \infty = 0$, then $0 > \hat{\beta}_i^T(t)$ is not satisfied. For this reason, we need to perform our analysis under the event $\{h_{i,t} \geqslant 1\}$.

The first thing to do is to remove the dependence on the number of pulls that, in a generic time instant, represents a random variable. So, we can write:

$$
\begin{aligned}
\mathbb{P}&\left( \exists t \in [\![T]\!] \,:\, |\hat{\mu}_i(N_{i,t}) - \bar{\mu}_i(N_{i,t})| > \hat{\beta}_i(N_{i,t}) \right) \\
&\leqslant \mathbb{P}\left( \exists n \in [\![0,T]\!] \text{ s.t. } h(n) \geqslant 1 \,:\, |\hat{\mu}_i(n) - \bar{\mu}_i(n)| > \hat{\beta}_i(n) \right) \\
&\leqslant \sum_{n \in [\![0,T]\!]:h(n) \geqslant 1} \mathbb{P}\left( |\hat{\mu}_i(n) - \bar{\mu}_i(n)| > \hat{\beta}_i(n) \right), \tag{22}
\end{aligned}
$$

where Equation 22 follows form a union bound over the possible values of $N_{i,t}$.

Now that we have a fixed value of $n$, consider a generic time $t$ in which arm $i$ has been pulled. We will observe a reward $x_t$ composed by the mean of the process $\mu_i(N_{i,t})$ plus some noise. The noise will be equal to $\eta_i(N_{i,t}) = x_t - \mu_i(N_{i,t})$, i.e., as the difference (not known) between the observed value for the arm $i$ at time $t$ and its real value at the same time. Let us rewrite the quantity to be bounded as follows, for every $n$:

$$h(n)\left(\hat{\mu}_i(n) - \bar{\mu}_i(n)\right) = \sum_{l=n-h(n)+1}^{n} \eta_i(l).$$

Here we can note that all the quantities $\eta_i(l)$ and $\eta_i(l - h(n))$ are independent since the number of pulls $l$ is fully determined by $n$ and $h(n)$, that now are non-random quantities.

Now, we apply the Azuma-Hoëffding's inequality of Lemma C.5 from Metelli et al. (2022) for sums of subgaussian martingale difference sequences. For a fixed $n$, we have:

$$\mathbb{P}\left(|\hat{\mu}_i(n) - \bar{\mu}_i(n)| \geqslant \hat{\beta}_i(n)\right) \leqslant \mathbb{P}\left(\left|\sum_{l=n-h(n)+1}^{n} \eta_i(l)\right| \geqslant h(n) \cdot \hat{\beta}_i^T(t)\right)$$

$$\leqslant 2\exp\left(-\frac{h(n)\hat{\beta}_i^T(n)^2}{2\sigma^2}\right)$$

$$= 2\exp\left(-\frac{a}{2}\right).$$

By replacing the obtained result into Equation 22 we get:

$$\sum_{n\in[\![0,T]\!]:h(n)\geqslant 1} 2\exp\left(-\frac{a}{2}\right) \leqslant \sum_{n=1}^{t} 2\exp\left(-\frac{a}{2}\right) \leqslant 2T\exp\left(-\frac{a}{2}\right).$$

$\square$

**Lemma 3.1** (Concentration of $\hat{\mu}_i$). *Under Assumption 2.1, for every $a > 0$, simultaneously for every arm $i \in [\![K]\!]$ and number of pulls $n \in [\![0,T]\!]$, with probability at least $1 - 2TKe^{-a/2}$ it holds that:*

$$\hat{\beta}_i(n) - \hat{\zeta}_i(n) \leqslant \hat{\mu}_i(n) - \mu_i(n) \leqslant \hat{\beta}_i(n),$$

*where $\hat{\beta}_i(n) := \sigma\sqrt{\frac{a}{h(n)}}$ and $\hat{\zeta}_i(n) := \frac{1}{2}(2T - n + h(n) - 1)\gamma_i(n - h(n) + 1)$.*

*Proof.* The proof simply combines Lemmas E.4 and E.5 and a union bound over the arms. $\square$

**Lemma 3.2** (Concentration of $\check{\mu}_i^T$). *Under Assumption 2.1, for every $a > 0$, simultaneously for every arm $i \in [\![K]\!]$ and number of pulls $n \in [\![0,T]\!]$, with probability at least $1 - 2TKe^{-a/10}$ it holds that:*

$$\check{\beta}_i^T(n) \leqslant \check{\mu}_i^T(n) - \mu_i(n) \leqslant \check{\beta}_i^T(n) + \check{\zeta}_i^T(n),$$

*where $\check{\beta}_i^T(n) := \sigma\cdot(T-n+h(n)-1)\sqrt{\frac{a}{h(n)^3}}$ and $\check{\zeta}_i^T(n) := \frac{1}{2}(2T-n+h(n)-1)\gamma_i(n-2h(n)+1)$.*

*Proof.* The proof simply combines Lemmas E.2 and E.3 and a union bound over the arms. $\square$

### E.2   Proofs of Section 4

In this appendix, we provide the proofs we have omitted in the main paper for what concerns the theoretical results about R-UCBE. All the lemma below are assuming that the strategy we use for selecting the arm is R-UCBE.

Let us define the *good event* $\Psi$ corresponding to the scenario in which all (over the rounds and over the arms) the bounds $B_i^T(n)$ hold for the projection up to time $T$ of the real reward expected value

$\mu_i(n)$, formally:

$$\Psi := \left\{ \forall i \in [\![K]\!], \forall t \in [\![T]\!] : |\check{\mu}_i^T(t) - \tilde{\mu}_i^T(t)| < \check{\beta}_i^T(t) \right\},$$

where $\tilde{\mu}_i^T(t)$ is the deterministic counterpart of $\check{\mu}_i^T(t)$ considering the expected payoffs $\mu_i(\cdot)$ instead of the realizations, formally:

$$\tilde{\mu}_i^T(N_{i,t}) := \frac{1}{h_{i,t}} \sum_{l=N_{i,t-1}-h_{i,t}+1}^{N_{i,t-1}} \left( \mu_i(l) + (T-l)\frac{\mu_i(l) - \mu_i(l-h_{i,t})}{h_{i,t}} \right).$$

**Lemma E.6.** *Under Assumption 2.1 and assuming that the good event $\Psi$ holds, the maximum number of pulls $N_{i,T}$ of a sub-optimal arm ($i \neq i^*(T)$) performed by the* R-UCBE *is upper bounded by the maximum integer $y_i(a)$ which satisfies the following condition:*

$$T\gamma_i(\lfloor(1-2\varepsilon)y_i(a)\rfloor) + 2T\sigma \cdot \sqrt{\frac{a}{\lfloor \varepsilon y_i(a) \rfloor^3}} \geqslant \Delta_i(T).$$

*Proof.* In the following, we will use $\tilde{\mu}_i^T(N_{i,t-1})$ to bound the bias introduced by $\check{\mu}_i^T(N_{i,t-1})$ and, subsequently, to find a number of pulls such that the algorithm cannot suggest pulling a suboptimal arm. Using Lemma E.4, we have that for every $i \in [\![K]\!]$ and $t \in [\![T]\!]$ and when $1 \leqslant h_{i,t} \leqslant \lfloor 1/2 \cdot N_{i,t-1} \rfloor$ with $N_{i,t-1} \geqslant 2$, it holds that:

$$\tilde{\mu}_i^T(N_{i,t-1}) - \mu_i(T) \leqslant \frac{1}{2} \cdot (2T - 2N_{i,t-1} + h_{i,t} - 1) \cdot \gamma_i(N_{i,t-1} - 2h_{i,t} + 1). \quad (23)$$

Let us assume that, at round $t$, the R-UCBE algorithm pulls the arm $i \in [\![K]\!]$ such that $i \neq i^*(T)$. From now on, to avoid weighing down the notation, we will omit the dependence of the optimal arm $i^*(T)$ on the budget $T$, simply denoting it as $i^*$, and the window size will be denoted as $h_{i,t} = h(N_{i,t-1})$. By construction, the algorithm chooses the arm with the largest upper confidence bound $B_i^T(N_{i,t-1})$. Thus, we have that $B_i^T(N_{i,t-1}) \geqslant B_{i^*}^T(N_{i^*,t-1})$. Now, we want to identify the minimum number of pulls such that this event no longer occurs, assuming that the good event $\Psi$ holds. We have that, if we pull such an arm $i \neq i^*$, it holds:

$$B_i^T(N_{i,t-1}) \geqslant B_{i^*}^T(N_{i^*,t-1})$$
$$B_i^T(N_{i,t-1}) - B_{i^*}^T(N_{i^*,t-1}) \geqslant 0$$
$$\Delta_i(T) + B_i^T(N_{i,t-1}) - B_{i^*}^T(N_{i^*,t-1}) \geqslant \Delta_i(T).$$

Using the definition of $\Delta_i(T)$ and the definition of the upper confidence bound $B_i^T(N_{i,t-1})$ in Equation 3 for $i$ and $i^*$, we have:

$$\mu_{i^*}(T) - \mu_i(T) + \check{\mu}_i^T(N_{i,t-1}) + \check{\beta}_i^T(N_{i,t-1}) - \check{\mu}_{i^*}^T(N_{i^*,t-1}) - \check{\beta}_{i^*}^T(N_{i^*,t-1}) \geqslant \Delta_i(T).$$

Given Assumption 2.1, we have that $\mu_{i^*}(T) \leqslant \tilde{\mu}_{i^*}^T(N_{i^*,t-1})$, and, therefore, we have:

$$\tilde{\mu}_{i^*}^T(N_{i^*,t-1}) - \mu_i(T) + \check{\mu}_i^T(N_{i,t-1}) + \check{\beta}_i^T(N_{i,t-1}) - \check{\mu}_{i^*}^T(N_{i^*,t-1}) - \check{\beta}_{i^*}^T(N_{i^*,t-1}) \geqslant \Delta_i(T),$$

and, since under the good event $\Psi$, it holds that $\tilde{\mu}_{i^*}^T(N_{i^*,t-1}) - \check{\mu}_{i^*}^T(N_{i^*,t-1}) - \check{\beta}_{i^*}^T(N_{i^*,t-1}) < 0$, we have:

$$-\mu_i(T) + \check{\mu}_i^T(N_{i,t-1}) + \check{\beta}_i^T(N_{i,t-1}) \geqslant \Delta_i(T)$$
$$-\mu_i(T) + \check{\beta}_i^T(N_{i,t-1}) + \tilde{\mu}_i^T(N_{i,t-1}) + \underbrace{\check{\mu}_i^T(N_{i,t-1}) - \tilde{\mu}_i^T(N_{i,t-1})}_{(D)} \geqslant \Delta_i(T),$$

where we added and subtracted $\tilde{\mu}_i^T(N_{i,t-1})$ in the last equation. Under the good event $\Psi$, we can upper bound $|(D)| = |\tilde{\mu}_i^T(N_{i,t-1}) - \check{\mu}_i^T(N_{i,t-1})| < \check{\beta}_i^T(N_{i,t-1})$, to get:

$$\tilde{\mu}_i^T(N_{i,t-1}) - \mu_i(T) + 2\check{\beta}_i^T(N_{i,t-1}) \geqslant \Delta_i(T).$$

Using Equation 23, and substituting the definition of $\check{\beta}_i^T(N_{i,t-1})$ provided in Equation 4, we have:

$$\frac{1}{2}\underbrace{(2T - 2N_{i,t-1} + h_{i,t} - 1)}_{\leqslant 2T}\cdot\gamma_i(N_{i,t-1} - 2h_{i,t} + 1)+$$

$$+2\sigma\cdot\underbrace{(T - N_{i,t-1} + h_{i,t} - 1)}_{\leqslant T}\cdot\sqrt{\frac{a}{h_{i,t}^3}} \geqslant \Delta_i(T)$$

$$\underbrace{T\cdot\gamma_i(\lfloor(1 - 2\varepsilon)N_{i,t}\rfloor)}_{(A)} + \underbrace{2\sigma T\sqrt{\frac{a}{\lfloor\varepsilon N_{i,t}\rfloor^3}}}_{(B)} \geqslant \Delta_i(T), \tag{24}$$

where we used the definition of $h_{i,t} := \lfloor\varepsilon N_{i,t}\rfloor$, observing that $N_{i,t-1} - 2\lfloor\varepsilon N_{i,t-1}\rfloor + 1 \geqslant (1 - 2\varepsilon)N_{i,t-1} + 1 = (1 - 2\varepsilon)N_{i,t} + 2\varepsilon \geqslant \lfloor(1 - 2\varepsilon)N_{i,t}\rfloor$ the fact that $N_{i,t-1} = N_{i,t} - 1$ since at time $t$ the algorithm pulls the $i$-th arm. This concludes the proof. $\square$

**Theorem 4.1.** *Let $\boldsymbol{\nu}$ be a rested MAB. Under Assumption 2.1, let $a^*$ be the largest positive value of $a$ satisfying:*

$$T - \sum_{i\neq i^*(T)} y_i(a) \geqslant 1, \tag{5}$$

*where for every $i \in [\![K]\!]$, $y_i(a)$ is the largest integer for which it holds:*

$$\underbrace{T\gamma_i(\lfloor(1 - 2\varepsilon)y\rfloor)}_{(A)} + \underbrace{2T\sigma\sqrt{\frac{a}{\lfloor\varepsilon y\rfloor^3}}}_{(B)} \geqslant \Delta_i(T). \tag{6}$$

*If $a^*$ exists, then for every $a \in [0, a^*]$ the error probability of $\mathtt{R-UCBE}$ is bounded by:*

$$e_T(\boldsymbol{\nu}, \mathtt{R-UCBE}) \leqslant 2TK\exp\left(-\frac{a}{10}\right). \tag{7}$$

*Proof.* From the definition of the error probability, we have:

$$e_T(\boldsymbol{\nu}, \mathtt{R-UCBE}) = \mathop{\mathbb{P}}_{\boldsymbol{\nu}, \mathtt{R-UCBE}}\left(\hat{I}^*(T) \neq i^*(T)\right) = \mathop{\mathbb{P}}_{\boldsymbol{\nu}, \mathtt{R-UCBE}}\left(I_{T+1} \neq i^*(T)\right).$$

Therefore, we need to evaluate the probability that the $\mathtt{R-UCBE}$ algorithm would pull a suboptimal arm in the $T + 1$ round. Given that Assumption 2.1 and that each suboptimal arm has been pulled a number of times $N_{i,T}$ (and cannot be pulled more) at the end of the time budget $T$, under the good event $\Psi$, we are guaranteed to recommend the optimal arm if:

$$T - \sum_{i\neq i^*(T)} N_{i,T} \geqslant 1. \tag{25}$$

If Equation 25 holds, a suboptimal arm can be selected by $\mathtt{R-UCBE}$ for the next round $T + 1$ only if the good event $\Psi$ does not hold $e_T(\boldsymbol{\nu}, \mathtt{R-UCBE}) = \mathbb{P}_{\boldsymbol{\nu}, \mathtt{R-UCBE}}(\Psi^c)$, where we denote with $\Psi^c$ the complementary of event $\Psi$. This probability is upper bounded by Lemma E.5 as:

$$e_T(\boldsymbol{\nu}, \mathtt{R-UCBE}) = \mathop{\mathbb{P}}_{\boldsymbol{\nu}, \mathtt{R-UCBE}}(\Psi^c) \leqslant 2TK\exp\left(-\frac{a}{10}\right).$$

We now derive a condition for $a$ in order to make Equation 25 hold. Thanks to Lemma E.6 we know that $N_{i,T} \leqslant y_i(a)$ where $y_i(a)$ is the maximum integer such that:

$$T\gamma_i(\lfloor(1 - 2\varepsilon)y_i(a)\rfloor) + 2T\sigma\sqrt{\frac{a}{\lfloor\varepsilon y_i(a)\rfloor^3}} \geqslant \Delta_i(T).$$

From this condition, we observe that $y_i(a)$ is an increasing function of $a$. Therefore, we can select $a$ in the interval $[0, a^*]$, where $a^*$ is the maximum value of $a$ such that:

$$T - \sum_{i \neq i*(T)} y_i(a) \geqslant 1. \tag{26}$$

Note that we are not guaranteed that such a value of $a^* > 0$ exists. In such a case, we cannot provide meaningful guarantees on the error probability of R-UCBE. $\qquad \square$

**Corollary 4.2.** *Let $\nu$ be a rested MAB. Under Assumptions 2.1 and 2.2, if the time budget $T$ satisfies:*

$$T \geqslant \begin{cases} \left(c^{\frac{1}{\beta}}(1 - 2\varepsilon)^{-1}\left(H_{1,1/\beta}(T)\right) + (K-1)\right)^{\frac{\beta}{\beta-1}} & \text{if } \beta \in (1, 3/2) \\ \left(c^{\frac{2}{3}}(1 - 2\varepsilon)^{-\frac{2}{3}\beta}\left(H_{1,2/3}(T)\right) + (K-1)\right)^{3} & \text{if } \beta \in [3/2, +\infty) \end{cases}, \tag{8}$$

*there exists $a^* > 0$ defined as:*

$$a^* = \begin{cases} \frac{\varepsilon^3}{4\sigma^2}\left(\left(\frac{T^{1-1/\beta}-(K-1)}{H_{1,1/\beta}(T)}\right)^{\beta} - c(1-2\varepsilon)^{-\beta}\right)^2 & \text{if } \beta \in (1, 3/2) \\ \frac{\varepsilon^3}{4\sigma^2}\left(\left(\frac{T^{1/3}-(K-1)}{H_{1,2/3}(T)}\right)^{3/2} - c(1-2\varepsilon)^{-\beta}\right)^2 & \text{if } \beta \in [3/2, +\infty) \end{cases}, \tag{9}$$

*where $H_{1,\eta}(T) := \sum_{i \neq i*(T)} \Delta_i(T)^{-\eta}$ for $\eta > 0$. Then, for every $a \in [0, a^*]$, the error probability of R-UCBE is bounded as in Equation 7.*

*Proof.* We recall that Assumption 2.2 states that all the increment functions $\gamma_i(n)$ are such that $\gamma_i(n) \leqslant cn^{-\beta}$. We use such a fact to provide an explicit solution for the optimal value of $a^*$. From Theorem 4.1 and using the fact that $\gamma_i(n) \leqslant cn^{-\beta}$, we have that Equation 6 can be rephrased as:

$$\frac{Tc}{\lfloor(1-2\varepsilon)y\rfloor^{\beta}} + \frac{2tT\sigma a^{\frac{1}{2}}}{\lfloor\varepsilon y\rfloor^{\frac{3}{2}}} \geqslant \Delta_i(T). \tag{27}$$

Or, more restrictively:

$$\frac{Tc(1-2\varepsilon)^{-\beta}}{(y-1)^{\beta}} + \frac{2T\sigma\varepsilon^{-\frac{3}{2}}a^{\frac{1}{2}}}{(y-1)^{\frac{3}{2}}} \geqslant \Delta_i(T).$$

Let us solve Equation 27 by analyzing separately the two cases in which one of the two terms in the l.h.s. of such equation becomes prevalent.

**Case 1:** $\beta \in \left[\frac{3}{2}, +\infty\right)$ In this branch, we can upper bound the left-side part of the inequality in Equation 27 by:

$$\frac{Tc(1-2\varepsilon)^{-\beta}}{(y-1)^{\frac{3}{2}}} + \frac{2T\sigma\varepsilon^{-\frac{3}{2}}a^{\frac{1}{2}}}{(y-1)^{\frac{3}{2}}} \geqslant \Delta_i(T).$$

Thus, we can derive:

$$y_i(a) \leqslant 1 + \left(\frac{Tc(1-2\varepsilon)^{-\beta} + 2T\sigma\varepsilon^{-\frac{3}{2}}a^{\frac{1}{2}}}{\Delta_i(T)}\right)^{\frac{2}{3}}. \tag{28}$$

Using the above value in Equation 26, provides:

$$T - \sum_{i \neq i*(T)} y_i(a) > 0$$

$$T - (K-1) - \left(Tc(1-2\varepsilon)^{-\beta} + 2T\sigma\varepsilon^{-\frac{3}{2}}a^{\frac{1}{2}}\right)^{\frac{2}{3}} \sum_{i \neq i*(T)} \frac{1}{\Delta_i^{\frac{2}{3}}(T)} > 0$$

$$T - (K-1) - \left(Tc\left(1-2\varepsilon\right)^{-\beta} + 2T\sigma\varepsilon^{-\frac{3}{2}}a^{\frac{1}{2}}\right)^{\frac{2}{3}} H_{1,2/3}(T) > 0$$

$$a < \frac{\left(\frac{(T^{1/3} - T^{-2/3}(K-1))^{\frac{3}{2}}}{(H_{1,2/3}(T))^{\frac{3}{2}}} - c(1-2\varepsilon)^{-\beta}\right)^2}{4\sigma^2\varepsilon^{-3}}$$

$$\implies a < \frac{\left(\frac{(T^{1/3} - (K-1))^{\frac{3}{2}}}{(H_{1,2/3}(T))^{\frac{3}{2}}} - c(1-2\varepsilon)^{-\beta}\right)^2}{4\sigma^2\varepsilon^{-3}},$$

where the last expression is obtained by observing that $T \geqslant 1$ and for obtaining a more manageable expression, under the assumption that $\frac{(T^{1/3} - (K-1))^{\frac{3}{2}}}{(H_{1,2/3}(T))^{\frac{3}{2}}} - c(1-2\varepsilon)^{-\beta} \geqslant 0$.

This implies a constraint on the minimum time budget $T$, whose explicit form for the case $\beta \in \left[\frac{3}{2}, +\infty\right)$ is provided in Lemma E.7.

**Case 2:** $\beta \in \left(1, \frac{3}{2}\right)$ In this case, we enforce the more restrictive condition:

$$\frac{Tc\left(1-2\varepsilon\right)^{-\beta}}{(y-1)^\beta} + \frac{2T\sigma\varepsilon^{-\frac{3}{2}}a^{\frac{1}{2}}}{(y-1)^\beta} \geqslant \Delta_i(T),$$

the value for the number of pulls is:

$$y_i(a) \leqslant 1 + \left(\frac{Tc\left(1-2\varepsilon\right)^{-\beta} + 2T\sigma\varepsilon^{-\frac{3}{2}}a^{\frac{1}{2}}}{\Delta_i(T)}\right)^{\frac{1}{\beta}}. \tag{29}$$

and the value for $a*$ becomes:

$$T - \sum_{i \neq i*(T)} N_{i,T} > 0$$

$$T - (K-1) - \left(Tc\left(1-2\varepsilon\right)^{-\beta} + 2T\sigma\varepsilon^{-\frac{3}{2}}a^{\frac{1}{2}}\right)^{\frac{1}{\beta}} \sum_{i \neq i*(T)} \frac{1}{\Delta_i^{\frac{1}{\beta}}(T)} > 0$$

$$T - (K-1) - \left(Tc\left(1-2\varepsilon\right)^{-\beta} + 2T\sigma\varepsilon^{-\frac{3}{2}}a^{\frac{1}{2}}\right)^{\frac{1}{\beta}} H_{1,1/\beta}(T) > 0$$

$$a < \frac{\left(\frac{(T^{1-1/\beta} - T^{-1/\beta}(K-1))^\beta}{(H_{1,1/\beta}(T))^\beta} - c(1-2\varepsilon)^{-\beta}\right)^2}{4\sigma^2\varepsilon^{-3}}$$

$$a < \frac{\left(\frac{(T^{1-1/\beta} - (K-1))^\beta}{(H_{1,1/\beta}(T))^\beta} - c(1-2\varepsilon)^{-\beta}\right)^2}{4\sigma^2\varepsilon^{-3}},$$

where the last expression is obtained by observing that $T \geqslant 1$ and for obtaining a more convenient expression, under the assumption that $\frac{(T^{1-1/\beta} - (K-1))^\beta}{(H_{1,1/\beta}(T))^\beta} - c(1-2\varepsilon)^{-\beta} \geqslant 0$.

Also here, this implies a constraint on the minimum time budget $T$ for the case $\beta \in \left(1, \frac{3}{2}\right)$, whose explicit form is provided in Lemma E.7 $\qquad\square$

**Lemma E.7.** *Under Assumptions 2.1 and 2.2, the minimum time budget $T$ for which the theoretical guarantees of* R−UCBE *hold is:*

$$T \geqslant \begin{cases} \left(c^{\frac{1}{\beta}}(1-2\varepsilon)^{-1}\left(H_{1,1/\beta}(T)\right) + (K-1)\right)^{\frac{\beta}{\beta-1}} & \text{if } \beta \in (1, 3/2) \\ \left(c^{\frac{2}{3}}(1-2\varepsilon)^{-\frac{2}{3}\beta}\left(H_{1,2/3}(T)\right) + (K-1)\right)^3 & \text{if } \beta \in [3/2, +\infty) \end{cases}.$$

*Proof.* Given Corollary 4.2, we want to find the values of $T$ such that $a* \geqslant 0$. This implies having $a* \geqslant 0$. Given the value of $\beta$, we can derive a lower bound for the time budget $T$.

**Case 1:** $\beta \in \left[\frac{3}{2}, +\infty\right)$ :

$$\frac{(T^{1/3} - (K-1))^{\frac{3}{2}}}{(H_{1,2/3}(T))^{\frac{3}{2}}} - c(1 - 2\varepsilon)^{-\beta} \geqslant 0.$$

From this, it follows that:

$$T \geqslant \left(c^{2/3}(1 - 2\varepsilon)^{-2/3 \cdot \beta} \left(H_{1,2/3}(T)\right) + (K-1)\right)^3.$$

**Case 2:** $\beta \in \left(1, \frac{3}{2}\right)$:

$$\frac{(T^{1-1/\beta} - (K-1))^{\beta}}{(H_{1,1/\beta}(T))^{\beta}} - c(1 - 2\varepsilon)^{-\beta} \geqslant 0.$$

From this, it follows that:

$$T \geqslant \left(c^{\frac{1}{\beta}}(1 - 2\varepsilon)^{-1} \left(H_{1,1/\beta}(T)\right) + (K-1)\right)^{\frac{\beta}{\beta-1}}.$$

$\square$

### E.3 PROOFS OF SECTION 5

In this appendix, we provide the proofs we have omitted in the main paper for what concerns the theoretical results about R-SR. We recall that with a slight abuse of notation, as done in Section 5, we denote with $\Delta_{(i)}(T)$ the $i^{\text{th}}$ gap rearranged in increasing order, i.e., we have $\Delta_{(i)}(T) \leqslant \Delta_{(j)}(T)$ for $2 \leqslant i < j \leqslant K$.

**Lemma E.8.** *For every arm $i \in [\![K]\!]$ and every round $t \in [\![T]\!]$, let us define:*

$$\bar{\mu}_i(t) = \frac{1}{h_{i,t}} \sum_{l=N_{i,t-1}-h_{i,t}+1}^{N_{i,t-1}} \mu_i(l),$$

*if $N_{i,t} \geqslant 2$, then $\mu_i(t) \geqslant \bar{\mu}_i(t)$, and if $h_{i,t} = \lfloor \varepsilon N_{i,t} \rfloor$ with $\varepsilon \in (0,1)$, it holds that:*

$$\mu_i(T) - \bar{\mu}_i(N_{i,t}) \leqslant T\gamma_i(\lceil (1-\varepsilon)N_{i,t} \rceil). \tag{30}$$

*Proof.* The proof follows from Lemma E.4, having observed that $N_{i,t-1} - \lfloor \varepsilon N_{i,t-1} \rfloor + 1 = \lceil (1-\varepsilon)N_{i,t-1} \rceil + 1 = \lceil (1-\varepsilon)(N_{i,t}-1) \rceil + 1 \geqslant \lceil (1-\varepsilon)N_{i,t} \rceil$ since $N_{i,t-1} = N_{i,t}-1$ and $\lceil x+y \rceil \leqslant \lceil x \rceil + \lceil y \rceil$ with $x = (1-\varepsilon)(N_{i,t}-1)$ and $y = 1 - \varepsilon$. $\square$

**Lemma E.9** (Lower Bound for the Time Budget for R-SR). *Under Assumptions 2.1 and 2.2, the R-SR algorithm is s.t. if the time budget $T$ satisfies:*

$$T \geqslant \max\left\{2K, c^{\frac{1}{\beta-1}}2^{\frac{1+\beta}{\beta-1}}(1-\varepsilon)^{-\frac{\beta}{\beta-1}}\overline{\log}(K)^{\frac{\beta}{\beta-1}} \max_{i \in [\![2,K]\!]} \left\{i\Delta_{(i)}^{-\frac{1}{\beta}}(T)\right\}^{\frac{\beta}{\beta-1}}\right\},$$

*then, it holds that:*

$$\forall j \in [\![K-1]\!]: \qquad T\gamma_{(1)}(\lceil (1-\varepsilon)N_j \rceil) \leqslant \frac{\Delta_{(K+1-j)}(T)}{2}.$$

*Proof.* First, using Assumption 2.2, we have that:

$$T\gamma_{(1)}(\lceil (1-\varepsilon)N_j \rceil) \leqslant cT(1-\varepsilon)^{-\beta}N_j^{-\beta}.$$

Substituting the definition of $N_j$ into the above equation, we get:

$$cT(1-\varepsilon)^{-\beta}N_j^{-\beta} = cT(1-\varepsilon)^{-\beta} \cdot \left(\frac{1}{\overline{\log}(K)} \cdot \frac{T-K}{K+1-j}\right)^{-\beta} \tag{31}$$

$$\leqslant cT(1-\varepsilon)^{-\beta} \cdot \left( \frac{T}{2\overline{\log}(K)(K+1-j)} \right)^{-\beta}, \tag{32}$$

where line (32) follows from $T \geqslant 2K$. Requiring that, for $j \in [\![1, K-1]\!]$, the maximum possible bias is lower than a fraction of the suboptimality gap of arm $K + 1 - j$, we obtain:

$$cT(1-\varepsilon)^{-\beta} N_j^{-\beta} \leqslant \frac{\Delta_{(K+1-j)}(T)}{2}$$

$$cT(1-\varepsilon)^{-\beta} \left( \frac{T}{2\overline{\log}(K)(K+1-j)} \right)^{-\beta} \leqslant \frac{\Delta_{(K+1-j)}(T)}{2}$$

$$T^{1-\beta} \leqslant \frac{\Delta_{(K+1-j)}(T) \cdot (1-\varepsilon)^{\beta}}{c 2^{1+\beta} \cdot \left( \overline{\log}(K)(K+1-j) \right)^{\beta}}$$

$$T \geqslant \frac{\Delta_{(K+1-j)}^{\frac{1}{1-\beta}}(T) \cdot (1-\varepsilon)^{\frac{\beta}{1-\beta}}}{c^{\frac{1}{1-\beta}} 2^{\frac{1+\beta}{1-\beta}} \cdot \left( \overline{\log}(K)(K+1-j) \right)^{\frac{\beta}{1-\beta}}}.$$

Requiring that the above condition holds for all the phases $j \in [\![K-1]\!]$ we have:

$$T \geqslant \max_{j \in [\![K-1]\!]} \left\{ \frac{\Delta_{(K+1-j)}^{\frac{1}{1-\beta}}(T) \cdot (1-\varepsilon)^{\frac{\beta}{1-\beta}}}{c^{\frac{1}{1-\beta}} 2^{\frac{1+\beta}{1-\beta}} \cdot \left( \overline{\log}(K) \cdot (K+1-j) \right)^{\frac{\beta}{1-\beta}}} \right\}$$

$$\geqslant c^{-\frac{1}{1-\beta}} 2^{-\frac{1+\beta}{1-\beta}} (1-\varepsilon)^{\frac{\beta}{1-\beta}} \overline{\log}(K)^{-\frac{\beta}{1-\beta}} \max_{j \in [\![K-1]\!]} \left\{ \left( \frac{\Delta_{(K+1-j)}(T)}{(K+1-j)^{\beta}} \right)^{\frac{1}{1-\beta}} \right\}$$

$$\geqslant c^{-\frac{1}{1-\beta}} 2^{-\frac{1+\beta}{1-\beta}} (1-\varepsilon)^{\frac{\beta}{1-\beta}} \overline{\log}(K)^{-\frac{\beta}{1-\beta}} \cdot \max_{j \in [\![K-1]\!]} \left\{ \left( (K+1-j)^{\beta} \Delta_{(K+1-j)}^{-1}(T) \right)^{\frac{1}{\beta-1}} \right\}$$

$$\geqslant c^{-\frac{1}{1-\beta}} 2^{-\frac{1+\beta}{1-\beta}} (1-\varepsilon)^{\frac{\beta}{1-\beta}} \overline{\log}(K)^{-\frac{\beta}{1-\beta}} \max_{i \in [\![2,K]\!]} \left\{ i^{\frac{\beta}{\beta-1}} \Delta_{(i)}^{-\frac{1}{\beta-1}}(T) \right\}.$$

$\square$

**Theorem 5.1.** *Let $\boldsymbol{\nu}$ be a rested MAB. Under Assumption 2.1 if the time budget $T$ satisfies:*

$$\forall j \in [\![K-1]\!]: \qquad T\gamma_{(1)}(\lceil (1-\varepsilon)N_j \rceil) \leqslant \frac{\Delta_{(K-j+1)}(T)}{2}, \tag{11}$$

*then, the error probability of* R-SR *is bounded by:*

$$e_T(\boldsymbol{\nu}, R\text{-}SR) \leqslant \frac{K(K-1)}{2} \exp\left( -\frac{\varepsilon}{8\sigma^2} \cdot \frac{T-K}{\overline{\log}(K)H_2(T)} \right), \tag{12}$$

*where $H_2(T) := \max_{i \in [\![K]\!]} \left\{ i \Delta_{(i)}(T)^{-2} \right\}$ and $\overline{\log}(K) = \frac{1}{2} + \sum_{i=2}^{K} \frac{1}{i}$.*

*Proof.* The R-SR recommends the wrong arm when at the end of some phase $j$ the optimal arm has a pessimistic estimator $\hat{\mu}_1(N_j)$ smaller than that of some suboptimal arm. Formally, we bound the following probability:

$$e_T(\boldsymbol{\nu}, R\text{-}SR) \leqslant \mathbb{P}_{\boldsymbol{\nu}, R\text{-}SR} \left( \exists j \in [\![K-1]\!] \, \exists i \in [\![K+1-j, K]\!] : \hat{\mu}_{(1)}(N_j) < \hat{\mu}_{(i)}(N_j) \right)$$

$$\leqslant \sum_{j=1}^{K-1} \mathbb{P}_{\boldsymbol{\nu}, R\text{-}SR} \left( \exists i \in [\![K+1-j, K]\!] : \hat{\mu}_{(1)}(N_j) < \hat{\mu}_{(i)}(N_j) \right)$$

$$\leqslant \sum_{j=1}^{K-1} \sum_{i=K+1-j}^{K} \mathbb{P}_{\boldsymbol{\nu}, R\text{-}SR} \left( \hat{\mu}_{(1)}(N_j) \leqslant \hat{\mu}_{(i)}(N_j) \right),$$

where we use a union bound over the phases and over the arms still in the available arm set $\mathcal{X}_{j-1}$ in each phase. Let us focus on $\mathbb{P}_{\boldsymbol{\nu},\text{R-SR}}\left(\hat{\mu}_{(1)}(N_j) \leqslant \hat{\mu}_{(i)}(N_j)\right)$. We have that the optimal arm has a smaller pessimistic estimator than the $i^{\text{th}}$ one when:

$$\hat{\mu}_{(i)}(N_j) \geqslant \hat{\mu}_{(1)}(N_j)$$

$$\hat{\mu}_{(i)}(N_j) - \hat{\mu}_{(1)}(N_j) \geqslant 0$$

$$\mu_{(1)}(T) - \hat{\mu}_{(1)}(N_j) + \hat{\mu}_{(i)}(N_j) - \mu_{(i)}(T) \geqslant \Delta_{(i)}(T) \tag{33}$$

$$\underbrace{\mu_{(1)}(T) - \bar{\mu}_{(1)}(N_j)}_{\leqslant T \cdot \gamma_{(1)}(\lceil (1-\varepsilon)N_j \rceil)} - \hat{\mu}_{(1)}(N_j) + \bar{\mu}_{(1)}(N_j) + \hat{\mu}_{(i)}(N_j) \underbrace{-\mu_{(i)}(T)}_{\leqslant -\bar{\mu}_{(i)}(N_j)} \geqslant \Delta_{(i)}(T) \tag{34}$$

$$-\hat{\mu}_{(1)}(N_j) + \bar{\mu}_{(1)}(N_j) + \hat{\mu}_{(i)}(N_j) - \bar{\mu}_{(i)}(N_j) \geqslant \Delta_{(i)}(T) - T \cdot \gamma_{(1)}((1-\varepsilon)N_j), \tag{35}$$

where we added $\pm\Delta_{(i)}(T)$ to derive Equation 33, and added $\pm\bar{\mu}_{(1)}(N_j)$ to derive Equation 34, we used the results in Lemma E.8 and the fact that the reward function is increasing. Since the time budget satisfies the hypothesis of the theorem, we have:

$$\forall j \in [\![K-1]\!]: \quad T\gamma_{(1)}(\lceil(1-\varepsilon)N_j\rceil) \leqslant \frac{\Delta_{(K-j+1)}(T)}{2} \implies T\gamma_{(1)}(\lceil(1-\varepsilon)N_j\rceil) \leqslant \frac{\Delta_{(i)}(T)}{2}, \tag{36}$$

since $\Delta_{(K-j+1)}(T) \leqslant \Delta_{(i)}(T)$ for all $i \in [\![K-j+1, K]\!]$. Substituting into Equation 35 the above, we have:

$$-\hat{\mu}_{(1)}(N_j) + \bar{\mu}_{(1)}(N_j) + \hat{\mu}_{(i)}(N_j) - \bar{\mu}_{(i)}(N_j) \geqslant \frac{\Delta_{(i)}(T)}{2},$$

and the error probability becomes:

$$e_T(\boldsymbol{\nu}, \text{R-SR}) \leqslant \sum_{j=1}^{K-1} \sum_{i=K+1-j}^{K} \mathbb{P}_{\boldsymbol{\nu},\text{R-SR}}\left(-\hat{\mu}_{(1)}(N_j) + \bar{\mu}_{(1)}(N_j) + \hat{\mu}_{(i)}(N_j) - \bar{\mu}_{(i)}(N_j) \geqslant \frac{\Delta_{(i)}(T)}{2}\right).$$

For the previous argument, we apply the Azuma-Hoëffding's inequality to the latter probability:

$$e_T(\boldsymbol{\nu}, \text{R-SR}) \leqslant \sum_{j=1}^{K-1} \sum_{i=K+1-j}^{K} \exp\left(-\frac{\varepsilon N_j \left(\frac{\Delta_{(i)}(T)}{2}\right)^2}{2\sigma^2}\right)$$

$$\leqslant \sum_{j=1}^{K-1} j \exp\left(-\frac{\varepsilon N_j}{8\sigma^2}\Delta_{(K+1-j)}^2\right).$$

Now, given that:

$$\frac{\varepsilon N_j}{8\sigma^2}\Delta_{(K+1-j)}^2 \geqslant \frac{\varepsilon}{8\sigma^2}\frac{T-K}{\overline{\log}(K)(K+1-j)\Delta_{(K+1-j)}^{-2}}$$

$$\geqslant \frac{\varepsilon}{8\sigma^2}\frac{T-K}{\overline{\log}(K)H_2(T)},$$

we finally derive the following:

$$e_T(\boldsymbol{\nu}, \text{R-SR}) \leqslant \frac{K(K-1)}{2}\exp\left(-\frac{\varepsilon}{8\sigma^2}\frac{T-K}{\overline{\log}(K)H_2(T)}\right),$$

which concludes the proof. $\qquad\square$

**Corollary 5.2.** *Let $\boldsymbol{\nu}$ be a rested MAB. Under Assumptions 2.1 and 2.2, if the time budget $T$ satisfies:*

$$T \geqslant \max\left\{2K, 2^{\frac{\beta+1}{\beta-1}}c^{\frac{1}{\beta-1}}(1-\varepsilon)^{-\frac{\beta}{\beta-1}}\overline{\log}(K)^{\frac{\beta}{\beta-1}}\max_{i\in[\![2,K]\!]}\left\{i\Delta_{(i)}(T)^{-\frac{1}{\beta}}\right\}^{\frac{\beta}{\beta-1}}\right\}, \tag{13}$$

*then, the error probability of* R-SR *is bounded as in Equation 12.*

*Proof.* The result follows from a direct combination of Theorem 5.2 and Lemma E.9. ☐

**Theorem E.10** (Simple regret of R-SR). *Let $\nu$ be a rested MAB and let $\mathfrak{A}$ be an algorithm. Let us define the expected simple regret as:*

$$r_T(\nu, \mathfrak{A}) := \mu_{i*(T)}(T) - \mathop{\mathbb{E}}_{\nu, \mathfrak{A}}[\mu_{\hat{I}*(T)}(N_{\hat{I}*(T),T})]. \tag{37}$$

*Under the same assumptions of Theorem 5.2, the expected simple regret suffered by* R-SR *is bounded by:*

$$r_T(\nu,\ R\text{-}SR) \leqslant \frac{K(K-1)}{2}\ \exp\left(-\frac{\varepsilon}{8\sigma^2} \cdot \frac{T-K}{\overline{\log}(K)H_2(T)}\right) + c4^\beta T^{1-\beta}\overline{\log}(K)^\beta$$

$$\leqslant \tilde{\mathcal{O}}\left(T^{1-\beta}\right).$$

*Proof.* Let us first observe that for the R-SR algorithm, we have:

$$r_T(\nu,\ \text{R-SR}) = \mu_{i*(T)}(T) - \mathop{\mathbb{E}}_{\nu,\ \text{R-SR}}[\mu_{\hat{I}*(T)}(N_{\hat{I}*(T),T})]$$

$$\leqslant \mathop{\mathbb{E}}_{\nu,\ \text{R-SR}}[\mathbb{1}\{\hat{I}*(T) \neq i^*(T)\} + (\mu_{i*(T)}(T) - \mu_{\hat{I}*(T)}(N_{\hat{I}*(T),T}))\mathbb{1}\{\hat{I}^*(T) = i^*(T)\}]$$

$$= \mathop{\mathbb{P}}_{\nu,\ \text{R-SR}}(\hat{I}^*(T) \neq i^*(T)) + \mathop{\mathbb{E}}_{\nu,\ \text{R-SR}}[(\mu_{i*(T)}(T) - \mu_{i*(T)}(N_{K-1}))\mathbb{1}\{\hat{I}^*(T) = i^*(T)\}]$$

$$\leqslant e_T(\nu,\ \text{R-SR}) + \mu_{i*(T)}(T) - \mu_{i*(T)}(N_{K-1}),$$

having observed that if the optimal arm is the one to be recommended, it had been pulled $N_{K-1}$ times. For the first term, we resort to the error probability in Equation 12, while for the second, we exploit Assumptions 2.1 and 2.2:

$$\mu_{i*(T)}(T) - \mu_{i*(T)}(N_{K-1}) \leqslant T\gamma_{i*(T)}(N_{K-1})$$

$$\leqslant cTN_{K-1}^{-\beta}$$

$$\leqslant cT\left(\left\lfloor\frac{1}{\overline{\log}(K)}\frac{T-K}{2}\right\rfloor\right)^{-\beta}$$

$$\leqslant c4^\beta T^{1-\beta}\overline{\log}(K)^\beta,$$

where the last inequality follows from $T \geqslant 2K$. Putting all together, we observe that the rate is of order $\tilde{\mathcal{O}}(e^{-T/H_2(T)} + T^{1-\beta}) = \tilde{\mathcal{O}}(T^{1-\beta})$. ☐

### E.4 PROOFS OF SECTION 6

In this appendix, we provide the proofs of the lower bound on the error probability presented in Section 6.

**Theorem 6.1.** *For every algorithm $\mathfrak{A}$, there exists a deterministic SRB $\nu$ satisfying Assumptions 2.1 and 2.2 with $c = \beta - 1$ and $K \geqslant 1 + 8^{1/(\beta-1)}$ such that the optimal arm $i^*(T)$ cannot be identified (i.e., $e_T(\nu, \mathfrak{A}) = 1$) for some time budgets $T$ unless:*

$$T \geqslant 8^{-\frac{1}{\beta-1}}H_{1,1/(\beta-1)}(T) = \sum_{i \neq i^*(T)}\left(\frac{1}{8\Delta_i(T)}\right)^{\frac{1}{\beta-1}}. \tag{14}$$

*Proof.* We define for every suboptimal arm $i \in [\![2, K]\!]$ the suboptimality gap reached at $T \to +\infty$ as $\Delta_i \in (0, 1/2]$. We consider the base instance $\nu$ (see Figure 5) in which define the (deterministic) reward functions are defined for $\beta > 1$ and $n \in \mathbb{N}$ as:

$$\mu_1(n) = \frac{1}{2}\left(1 - n^{1-\beta}\right),$$

$$\mu_i(n) = \min \left\{ \underbrace{\left( \frac{1}{2} + \Delta_i \right) \left( 1 - n^{1-\beta} \right)}_{=:\mu_i'(n)}, \frac{1}{2} - \Delta_i \right\} \qquad i \in [\![2, K]\!].$$

Clearly, $\nu$ fulfills Assumption 2.1 and it is simple to show that also Assumption 2.2 is satisfied. Indeed, by first-order Taylor expansion:

$$\gamma_1(n) = \mu_1(n+1) - \mu_1(n) \leqslant \sup_{x \in [n, n+1]} \frac{\partial}{\partial x} \mu_1(x) \tag{38}$$

$$= \frac{\beta - 1}{2} \sup_{x \in [n, n+1]} x^{-\beta} = \frac{\beta - 1}{2} n^{-\beta},$$

$$\gamma_i(n) = \mu_i(n+1) - \mu_i(n) \leqslant \sup_{x \in [n, n+1]} \frac{\partial}{\partial x} \mu_i'(x)$$

$$= (\beta - 1) \left( \frac{1}{2} + \Delta_i \right) \sup_{x \in [n, n+1]} x^{-\beta} = (\beta - 1) n^{-\beta}$$

Thus, we simply take $c = \beta - 1$ in Assumption 2.2. Let us define $n_i^*$ the number of pulls in which arm $i \in [\![2, K]\!]$ reaches the stationary behavior:

$$\left( \frac{1}{2} + \Delta_i \right) \left( 1 - n^{1-\beta} \right) = \frac{1}{2} - \Delta_i \implies n_i^* = \left( \frac{1/2 + \Delta_i}{2\Delta_i} \right)^{\frac{1}{\beta - 1}}.$$

A sufficient condition on the time budget so that the optimal arm is 1 (i.e., $i^*(T) = 1$) is given by $T \geqslant T^*$, where $T^*$ is the point in which the curve of the optimal arm intersects that of any of the suboptimal arms $i \in [\![2, K]\!]$:

$$\frac{1}{2}(1 - T^{1-\beta}) = \frac{1}{2} - \Delta_i \implies T^* := \max_{i \in [\![2, K]\!]} \left( \frac{1}{2\Delta_i} \right)^{\frac{1}{\beta - 1}}.$$

Consider now the regime in which $T \geqslant T^*$. We proceed by contradiction. Suppose that there exists an algorithm $\mathfrak{A}$ that identifies the optimal arm such that on the bandit $\nu$ the suboptimal arm $\bar{i} \in [\![2, K]\!]$ has an expected number of pulls satisfying:

$$\mathop{\mathbb{E}}_{\nu, \mathfrak{A}} [N_{\bar{i}, T}] < n_{\bar{i}}^*. \tag{39}$$

Consider now the alternative bandit $\nu'$ constructed from $\nu$ by keeping all the arms unaltered, except for arm $\bar{i}$ that is made optimal, defined for $n \in \mathbb{N}$:

$$\mu_{\bar{i}}'(n) = \left( \frac{1}{2} + \Delta_{\bar{i}} \right) \left( 1 - n^{1-\beta} \right),$$

$$\mu_j'(n) = \mu_j(n), \qquad j \in [\![K]\!] \backslash \{\bar{i}\}.$$

Clearly the bandit $\nu'$ fulfills Assumption 2.1 and, with calculations similar to those in Equation 38, we conclude that it satisfies Assumption 2.2 with $c = \beta - 1$. A sufficient condition on $T$ for which arm $\bar{i}$ is optimal in bandit $\nu'$ is that $T \geqslant T_2$ in which the curve of arm $\bar{i}$ intersects that of the arms $j$ such that $\Delta_j \geqslant \Delta_{\bar{i}}$:

$$\left( \frac{1}{2} + \Delta_{\bar{i}} \right) \left( 1 - T^{1-\beta} \right) = \frac{1}{2} - \Delta_j \implies T \geqslant \max_{j \in [\![K]\!] : \Delta_j \geqslant \Delta_{\bar{i}}} \left( \frac{1/2 + \Delta_{\bar{i}}}{\Delta_{\bar{i}} + \Delta_j} \right)^{\frac{1}{\beta - 1}}.$$

Thus, we take:

$$T_2 := \left( \frac{1/2 + \Delta_{\bar{i}}}{2\Delta_{\bar{i}}} \right)^{\frac{1}{\beta - 1}}.$$

Clearly, $T^* \geqslant T_2$ since all the suboptimality gaps are at most $1/2$. Thus, we continue in the regime $T \geqslant T^*$. Since $\mu'_{\bar{i}}(n) = \mu_{\bar{i}}(n)$ if $n < n^*_i$, it follows that under condition 39, algorithm $\mathfrak{A}$ cannot distinguish between the two bandits and, consequently, cannot identify the optimal arm on bandit $\nu'$. Thus, it must follow, from the contradiction, that:

$$\mathbb{E}_{\nu,\mathfrak{A}}[N_{\bar{i},T}] \geqslant n^*_{\bar{i}}.$$

By summing over $\bar{i} \in [\![2, K]\!]$, we obtain:

$$T \geqslant \sum_{\bar{i} \in [\![2,K]\!]} \mathbb{E}_{\nu}[N_{\bar{i}}(T)] \geqslant \sum_{\bar{i} \in [\![2,K]\!]} n^*_{\bar{i}} = \sum_{\bar{i} \in [\![2,K]\!]} \left(\frac{1/2 + \Delta_{\bar{i}}}{2\Delta_{\bar{i}}}\right)^{\frac{1}{\beta-1}} \geqslant \sum_{\bar{i} \in [\![2,K]\!]} \left(\frac{1}{4\Delta_{\bar{i}}}\right)^{\frac{1}{\beta-1}} =: T^{\dagger}.$$

(40)

Thus, we have found an interval $T \in [T^*, T^{\dagger}]$ in which identification cannot be performed. Notice that it is simple to enforce that $T^{\dagger} > T^*$. For instance, by taking all suboptimality gaps equal $\Delta_2 = \cdots = \Delta_K =: \Delta$, we have:

$$T^{\dagger} = (K-1)\left(\frac{1}{4\Delta}\right)^{\frac{1}{\beta-1}} \geqslant \left(\frac{1}{2\Delta}\right)^{\frac{1}{\beta-1}} = T^* \implies K \geqslant 1 + 2^{\frac{1}{\beta-1}}.$$

(41)

To conclude, we need to relate $\Delta_i$ with $\Delta_i(T)$ and $\Delta'_i(T)$. We perform the computation for both the instances $\nu$ and $\nu'$, in the regime $T \geqslant 2^{\frac{1}{\beta-1}}T^*$. Let us start with $\nu$:

$$\Delta_i(T) = \frac{1}{2}(1 - T^{1-\beta}) - \left(\frac{1}{2} - \Delta_i\right)$$

$$= \Delta_i - \frac{1}{2}T^{1-\beta}$$

$$\geqslant \Delta_i - \frac{1}{2}\min_{j \in [\![2,K]\!]}\Delta_j \geqslant \frac{\Delta_i}{2}, \quad i \in [\![2, K]\!]$$

We move to $\nu'$:

$$\Delta'_1(T) = \left(\frac{1}{2} + \Delta_{\bar{i}}\right)(1 - T^{1-\beta}) - \frac{1}{2}(1 - T^{1-\beta})$$

$$= \Delta_{\bar{i}}(1 - T^{1-\beta}) \geqslant \frac{\Delta_{\bar{i}}}{2}$$

$$= \Delta_{\bar{i}} - \Delta_{\bar{i}}\min_{j \in [\![2,K]\!]}\Delta_j$$

$$\geqslant \Delta_{\bar{i}} - \frac{1}{2}\Delta_{\bar{i}} = \frac{\Delta_{\bar{i}}}{2},$$

having recalled that all suboptimality gaps are smaller than $1/2$. For suboptimal arms, we have:

$$\Delta'_i(T) = \left(\frac{1}{2} + \Delta_{\bar{i}}\right)(1 - T^{1-\beta}) - \left(\frac{1}{2} - \Delta_i\right) \geqslant \Delta_i - \frac{1}{2}T^{1-\beta} \geqslant \frac{\Delta_i}{2}, \quad i \in [\![2,K]\!]\backslash\{\bar{i}\}.$$

Thus, a necessary condition for the correct identification of the optimal arm is:

$$T \geqslant \sum_{\bar{i} \in [\![2,K]\!]} \left(\frac{1/2 + 2\Delta_{\bar{i}}(T)}{4\Delta_{\bar{i}}(T)}\right)^{\frac{1}{\beta-1}} \geqslant \sum_{\bar{i} \in [\![2,K]\!]} \left(\frac{1}{8\Delta_{\bar{i}}(T)}\right)^{\frac{1}{\beta-1}} = 2^{-\frac{1}{\beta-1}}T^{\dagger}.$$

With a derivation similar to that of Equation 41, we have that for $K \geqslant 1 + 8^{\frac{1}{\beta-1}}$, we can enforce $2^{-\frac{1}{\beta-1}}T^{\dagger} \geqslant 2^{\frac{1}{\beta-1}}T^*$. $\qquad\square$

**Theorem 6.2.** *For every algorithm $\mathfrak{A}$ run with a time budget $T$ fulfilling Equation 14:*

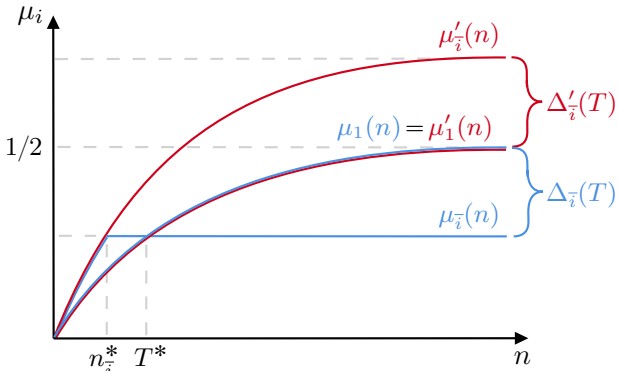

Figure 5: Instances $\boldsymbol{\nu}$ and $\boldsymbol{\nu}'$ of SRB used in Theorem 6.1 and Theorem 6.2.

- *there exists a set $\mathcal{V}_1$ of SRBs satisfying Assumptions 2.1 and 2.2 with $K \geqslant 8^{1/(\beta-1)}$ and bounded complexity index $H_{1,2}(T) := \sum_{i \neq i^*(T)} \Delta_i(T)^{-2} \leqslant \overline{H} < +\infty$ such that:*

$$\sup_{\boldsymbol{\nu} \in \mathcal{V}_1} e_T(\boldsymbol{\nu}, \mathfrak{A}) \geqslant \frac{1}{4} \exp\left(-\frac{8T}{\sigma^2 \overline{H}}\right);$$

- *there exists a set $\mathcal{V}_2$ of SRBs satisfying Assumptions 2.1 and 2.2 with $K \geqslant 8^{1/(\beta-1)}$, complexity indexes $H_{1,2}(T) \leqslant \overline{H}$, and $\overline{H} \geqslant 16K^2$ such that:*

$$\sup_{\boldsymbol{\nu} \in \mathcal{V}_2} e_T(\boldsymbol{\nu}, \mathfrak{A}) \exp\left(\frac{32T}{\sigma^2 \log(K) H_{1,2}(T)}\right) \geqslant \frac{1}{4}.$$

*Proof.* The proof combines the techniques of Kaufmann et al. (2016) with the ones of Carpentier & Locatelli (2016). We consider the Gaussian bandits with variance $\sigma^2$ equal for all the arms and the expected reward that are built in terms of the following prototypical instance for all $n \in [\![T]\!]$ and $k \in [\![2, K]\!]$:

$$\mu_1(n) = \frac{1}{2}\left(1 - n^{1-\beta}\right),$$

$$\mu_k(n) = \min\left\{\left(\frac{1}{2} + \Delta_k\right)(1 - n^{1-\beta}), \frac{1}{2} - \Delta_k\right\},$$

where $\Delta_k \in [0, 1/2]$ will be chosen later in the proof and $\Delta_1 = 0$. We have already shown in the proof of Theorem 6.1 that such instances fulfill both Assumptions 2.1 and 2.2. Consider now the instances $\boldsymbol{\nu}^{(i)}$ defined for $i \in [\![K]\!]$:

$$\mu_k^{(i)}(n) = \begin{cases} \mu_k(n) & \text{if } k \neq i \\ \left(\frac{1}{2} + \Delta_i\right)(1 - n^{1-\beta}) & \text{if } k = i \end{cases}. \tag{42}$$

For $T$ sufficiently large, whose lower bound is provided in Theorem 6.1, in bandit $\boldsymbol{\nu}^{(i)}$ the optimal arm is $i$. For all these instances, we can provide the asymptotic value of the expected rewards:

$$\mu_k^{(i)}(\infty) := \lim_{T \to +\infty} \mu_k^{(i)}(T) = \begin{cases} \frac{1}{2} - \Delta_k & \text{if } k \neq i \\ \frac{1}{2} + \Delta_i & \text{if } k = i \end{cases}. \tag{43}$$

Let us now define the following quantities for $i, k \in [\![K]\!]$:

$$\Delta_k^{(i)} := \mu_i^{(i)}(\infty) - \mu_k^{(i)}(\infty) = \begin{cases} 0 & \text{if } k = i \\ \Delta_i + \Delta_k & \text{if } k \neq i \end{cases}. \tag{44}$$

Thus, $\Delta_k^{(i)}$ represents the asymptotic sub-optimality gaps for bandit $\boldsymbol{\nu}^{(i)}$. Let us also define the complexity terms for $i \in [\![K]\!]$:

$$H_2^{(i)} := \sum_{k \neq i} \frac{1}{(\Delta_k^{(i)})^2}. \tag{45}$$

Furthermore, using an argument analogous to that of the proof of Theorem 6.1, we have that, for sufficiently large $T$ (whose value is provided in the statement of the theorem), $\Delta_k^{(i)} \leqslant 2\Delta_k^{(i)}(T)$, where $\Delta_k^{(i)}(T) := \mu_i^{(i)}(T) - \mu_k^{(i)}(T)$. Thus, we have that: $H_2^{(i)} \geqslant \frac{1}{4} H_{1,2}^{(i)}(T) := \sum_{k \neq i} \frac{1}{(\Delta_k^{(i)}(T))^2}$. Finally, we define:

$$H^* := \sum_{k \in [\![2,K]\!]} \frac{1}{\Delta_k^2 H_2^{(k)}}.$$

**Proof of the first statement:** Let $\mathfrak{A}$ be an algorithm, we immediately deduce that there exists an arm $\bar{i} \in [\![2, K]\!]$ such that:

$$\mathbb{E}_{\boldsymbol{\nu}^{(1)}, \mathfrak{A}}[N_{\bar{i}, T}] \leqslant \frac{T}{\Delta_{\bar{i}}^2 H_2^{(1)}}. \tag{46}$$

Indeed, if this is not the case, we have:

$$\sum_{\bar{i} \in [\![2,K]\!]} \mathbb{E}_{\boldsymbol{\nu}^{(1)}, \mathfrak{A}}[N_{\bar{i}, T}] > \sum_{\bar{i} \in [\![2,K]\!]} \frac{T}{\Delta_{\bar{i}}^2 H_2^{(1)}} = T,$$

having used the definition of $H_2^{(1)}$, leading to a contradiction. Thus, by using Bretagnolle-Huber's inequality, we obtain:

$$\max_{i \in [\![K]\!]} \{e_T(\boldsymbol{\nu}^{(i)})\} \geqslant \max\{e_T(\boldsymbol{\nu}^{(1)}), e_T(\boldsymbol{\nu}^{(\bar{i})})\}$$

$$\geqslant \frac{1}{4} \exp\left(- \mathbb{E}_{\boldsymbol{\nu}^{(1)}, \mathfrak{A}}\left[\sum_{t=1}^T \mathbb{1}\{I_t = \bar{i}\} D_{KL}\left(\nu_{\bar{i}}^{(1)}(N_{\bar{i},t}), \nu_{\bar{i}}^{(\bar{i})}(N_{\bar{i},t})\right)\right]\right)$$

$$= \frac{1}{4} \exp\left(- \mathbb{E}_{\boldsymbol{\nu}^{(1)}, \mathfrak{A}}\left[\sum_{t=1}^T \mathbb{1}\{I_t = \bar{i}\} \frac{(\mu_{\bar{i}}^{(1)}(N_{\bar{i},t}) - \mu_{\bar{i}}^{(\bar{i})}(N_{\bar{i},t}))^2}{2\sigma^2}\right]\right)$$

$$\geqslant \frac{1}{4} \exp\left(- \mathbb{E}_{\boldsymbol{\nu}^{(1)}, \mathfrak{A}}\left[N_{\bar{i},T}\right] \frac{(2\Delta_{\bar{i}})^2}{2\sigma^2}\right)$$

$$\geqslant \frac{1}{4} \exp\left(-\frac{2}{\sigma^2} \cdot \frac{T}{H_2^{(1)}}\right),$$

where we observed that for every $n \in [\![T]\!]$, we have:

$$|\mu_{\bar{i}}^{(1)}(N_{\bar{i},t}) - \mu_{\bar{i}}^{(\bar{i})}(N_{\bar{i},t})| \leqslant \max_{n \in [\![0,T]\!]} |\mu_{\bar{i}}^{(1)}(n) - \mu_{\bar{i}}^{(\bar{i})}(n)| \leqslant 2\Delta_{\bar{i}}.$$

This concludes the proof of the first statement, having observed that $H_2^{(1)} = \max_{k \in [\![K]\!]} H_2^{(k)}$.

**Proof of the second statement** Let $\mathfrak{A}$ be an algorithm, we immediately deduce that there exists an arm $\bar{i} \in [\![2, K]\!]$ such that:

$$\mathbb{E}_{\boldsymbol{\nu}^{(1)}, \mathfrak{A}}[N_{\bar{i}, T}] \leqslant \frac{T}{H^* \Delta_{\bar{i}}^2 H_2^{(\bar{i})}}. \tag{47}$$

Indeed, if this is not the case, we have:

$$\sum_{\bar{i}\in[\![2,K]\!]} \mathbb{E}_{\boldsymbol{\nu}^{(1)},\mathfrak{A}}[N_{\bar{i},T}] > \sum_{\bar{i}\in[\![2,K]\!]} \frac{T}{H^*\Delta_{\bar{i}}^2 H_2^{(\bar{i})}} = T,$$

having used the definition of $H^*$, leading to a contradiction. Thus, by using Bretagnolle-Huber's inequality, we obtain:

$$\max_{i\in[\![K]\!]}\{e_T(\boldsymbol{\nu}^{(i)})\} \geqslant \max\{e_T(\boldsymbol{\nu}^{(1)}), e_T(\boldsymbol{\nu}^{(\bar{i})})\}$$

$$\geqslant \frac{1}{4}\exp\left(-\mathbb{E}_{\boldsymbol{\nu}^{(1)},\mathfrak{A}}\left[\sum_{t=1}^{T} \mathbb{1}\{I_t = \bar{i}\} D_{KL}\left(\nu_{\bar{i}}^{(1)}(N_{\bar{i},t}), \nu_{\bar{i}}^{(\bar{i})}(N_{\bar{i},t})\right)\right]\right)$$

$$= \frac{1}{4}\exp\left(-\mathbb{E}_{\boldsymbol{\nu}^{(1)},\mathfrak{A}}\left[\sum_{t=1}^{T} \mathbb{1}\{I_t = \bar{i}\} \frac{(\mu_{\bar{i}}^{(1)}(N_{\bar{i},t}) - \mu_{\bar{i}}^{(\bar{i})}(N_{\bar{i},t}))^2}{2\sigma^2}\right]\right)$$

$$\geqslant \frac{1}{4}\exp\left(-\mathbb{E}_{\boldsymbol{\nu}^{(1)},\mathfrak{A}}\left[N_{\bar{i},T}\right]\frac{(2\Delta_{\bar{i}})^2}{2\sigma^2}\right)$$

$$\geqslant \frac{1}{4}\exp\left(-\frac{2}{\sigma^2} \cdot \frac{T}{H^* H_2^{(i)}}\right),$$

where we observed that for every $n \in [\![T]\!]$, we have $|\mu_{\bar{i}}^{(1)}(N_{\bar{i},t}) - \mu_{\bar{i}}^{(\bar{i})}(N_{\bar{i},t})| \leqslant 2\Delta_{\bar{i}}$. We now set the values of $\Delta_k$ so that to maximize the error probability. Specifically, similarly to Carpentier & Locatelli (2016), we chose $\Delta_k = k/(4K) \leqslant 1/2$ for $k \in [\![2, K]\!]$. Recalling the definitions of $H^*$ and $H_2^{(k)}$, we have for $k \in [\![2, K]\!]$:

$$\Delta_k^2 H_2^{(k)} = \Delta_k^2 \sum_{i\neq k} \frac{1}{(\Delta_i^{(k)})^2} = \Delta_k^2 \sum_{i\neq k} \frac{1}{(\Delta_i + \Delta_k)^2} = \sum_{i\neq k} \frac{k^2}{(i+k)^2}.$$

With simple algebraic bounds, we obtain:

$$\sum_{i\neq k} \frac{k^2}{(i+k)^2} \leqslant \sum_{1\leqslant <k} \frac{k^2}{k^2} + \sum_{k<i\leqslant K} \frac{k^2}{i^2} \leqslant k + k^2 \int_{x=k}^{K} \frac{1}{x^2}\mathrm{d}x = k + k^2\left(\frac{1}{k} - \frac{1}{K}\right) \leqslant 2k.$$

From this, recalling that $K \geqslant 2$, we have:

$$H^* = \sum_{k=2}^{K} \frac{1}{\Delta_k^2 H_2^{(k)}} \geqslant \sum_{k=2}^{K} \frac{1}{2k} \geqslant \frac{1}{2}\int_{x=2}^{K+1} \frac{1}{x}\mathrm{d}x \geqslant \frac{1}{2}\log\left(\frac{K+1}{2}\right) \geqslant \frac{1}{4}\log K.$$

We observe that $H_2^{(1)} = 16K^2 \sum_{k\in[\![2,K]\!]} \frac{1}{k^2} \leqslant 16K^2 \int_{x=1}^{K} \frac{1}{x^2}\mathrm{d}x = 16K^2\left(1 - \frac{1}{K}\right) \leqslant 16K^2.$ $\qquad\square$

### E.5 AUXILIARY LEMMAS

**Lemma E.11** (Höeffding-Azuma's inequality for weighted martingales). *Let $\mathcal{F}_1 \subset \cdots \subset \mathcal{F}_n$ be a filtration and $X_1, \ldots, X_n$ be real random variables such that $X_t$ is $\mathcal{F}_t$-measurable, $\mathbb{E}[X_t|\mathcal{F}_{t-1}] = 0$ (i.e., a martingale difference sequence), and $\mathbb{E}[\exp(\lambda X_t)|\mathcal{F}_{t-1}] \leqslant \exp\left(\frac{\lambda^2\sigma^2}{2}\right)$ for any $\lambda > 0$ (i.e., $\sigma^2$-subgaussian). Let $\alpha_1, \ldots, \alpha_n$ be non-negative real numbers. Then, for every $\kappa \geqslant 0$ it holds that:*

$$\mathbb{P}\left(\left|\sum_{t=1}^{n} \alpha_t X_t\right| > \kappa\right) \leqslant 2\exp\left(-\frac{\kappa^2}{2\sigma^2 \sum_{t=1}^{n} \alpha_t^2}\right).$$

*Proof.* For a complete proof of this statement, we refer to Lemma C.5 of Metelli et al. (2022). $\qquad\square$

**Lemma E.12.** *Let $\beta > 1$, then it holds that:*

$$H_{1,1/\beta}(T)^{\beta/(\beta-1)} \geqslant H_{1,1/(\beta-1)}(T).$$

*Proof.* We prove the equivalent statement, being $\beta > 1$:

$$H_{1,1/(\beta-1)}(T)^{(\beta-1)/\beta} \leqslant H_{1,1/\beta}(T).$$

Recalling that the function $(\cdot)^{(\beta-1)/\beta}$ is subadditive, being $(\beta-1)/\beta < 1$, we have:

$$H_{1,1/(\beta-1)}(T)^{(\beta-1)/\beta} = \left( \sum_{i \neq i^*(T)} \frac{1}{\Delta_i(T)^{1/(\beta-1)}} \right)^{(\beta-1)/\beta} \leqslant \sum_{i \neq i^*(T)} \frac{1}{\Delta_i(T)^{1/\beta}} = H_{1,1/\beta}(T).$$

$\square$

**Lemma E.13.** *(Adapted from (Audibert et al., 2010, Section 6.1)) Let $\eta > 1$, then it holds that:*

$$H_{2,\eta}(T) \leqslant H_{1,\eta}(T) \leqslant \overline{\log}(K)H_{2,\eta}(T),$$

*where:*

$$H_{2,\eta}(T) := \max_{i \in [\![2,K]\!]} \left\{ i\Delta_{(i)}^{-\eta}(T) \right\},$$

*and by convention $H_{2,2}(T) =: H_2(T)$.*

*Proof.* We simply apply the inequalities of (Audibert et al., 2010, Section 6.1) replacing $\Delta_{(i)}(T)$ with $\Delta_{(i)}^{\eta}(T)$. $\square$

## F THEORETICAL ANALYSIS OF A BASELINE: RR-SW

In this appendix, we provide the theoretical analysis for the algorithm Round Robin Sliding Window (RR-SW), as it represents the most intuitive baseline for this setting. First, we need to formalize the algorithm, whose pseudo-code is provided in Algorithm 3.

---

**Algorithm 3:** RR-SW.

---
**Input :** Time budget $T$, Number of arms $K$,
    Window size $\varepsilon$

1   Initialize $t \leftarrow 1$
2   Estimate $N = \frac{T}{K}$
3   **for** $i \in [\![K]\!]$ **do**
4       **for** $l \in [\![N]\!]$ **do**
5           Pull arm $i$ and observe $x_t$
6           $t \leftarrow t + 1$
7       **end**
8       Update $\hat{\mu}_i(N)$
9   **end**
10  Recommend $\widehat{I}^*(T) \in \arg\max_{i \in [\![K]\!]} \hat{\mu}_i(N)$

---

**Algorithm** The algorithm takes as input the time budget $T$, the number of arms $K$, and the window size $\varepsilon$. Then, it computes the number of pulls $N = \frac{T}{K}$ we need to perform for each arm. After having computed the number of pulls, RR-SW plays all the arms $N$ times in a round-robin fashion. After the $N$ pulls, it estimates $\hat{\mu}_i(N)$ using the last $\lfloor \varepsilon N \rfloor$ samples (i.e., the ones from $\lceil (1-\varepsilon)N \rceil$ to $N$). Finally, it recommends $\hat{I}^*(T)$, corresponding to the one which the highest estimated $\hat{\mu}_i(N)$.

**Error probability bound** Given this quantity, The error probability for the RR-SW algorithm can be bounded as follows.

**Theorem F.1.** *Under Assumptions 2.1 and 2.2, considering a time budget $T$ satisfying:*

$$T \geqslant 2^{\frac{1}{\beta-1}} \, c^{\frac{1}{\beta-1}} (1-\varepsilon)^{-\frac{\beta}{\beta-1}} \, K^{\frac{\beta}{\beta-1}} \, \Delta_{(2)}^{-\frac{1}{\beta-1}}(T), \tag{48}$$

*the error probability of* RR-SW *is bounded by:*

$$e_T(\boldsymbol{\nu}, RR\text{-}SW) \leqslant (K-1) \, \exp\left( -\frac{\varepsilon \, T}{8 \, K \, \sigma^2} \, \Delta_{(2)}^2(T) \right).$$

Some comments are in order. First, it is worth noting how, as expected, by increasing the number of samples considered in the estimator, we reduce the error probability $e_T$ at the cost of a stricter constraint on the time budget $T$. This is due to the request that the arms must be already separated at the beginning of the window we use to estimate the $\hat{\mu}_i(N)$. Second, the error probability scales as an (inverse) function only of the smallest suboptimality gap $\Delta_{(2)}(T)$ and no longer on the (more convenient) complexity index $H_2(T)$.

## F.1 PROOFS

Before demonstrating Theorem F.1, we need to introduce the following technical lemma.

**Lemma F.2** (Lower Bound for the Time Budget). *Under Assumptions 2.1 and 2.2, the* RR-SW *algorithm is s.t. the minimum value for the horizon $T$ ensuring that:*

$$T \, \gamma_1(\lceil (1-\varepsilon)N \rceil) \leqslant \frac{\Delta_{(2)}(T)}{2},$$

*where $N = \frac{T}{K}$ and $\varepsilon \in (0,1)$ is:*

$$T \geqslant 2^{\frac{1}{\beta-1}} \, c^{\frac{1}{\beta-1}} (1-\varepsilon)^{-\frac{\beta}{\beta-1}} \, K^{\frac{\beta}{\beta-1}} \, \Delta_{(2)}^{-\frac{1}{\beta-1}}(T).$$

*Proof.* Given that, for Assumptions 2.1 and 2.2, it holds:

$$T\gamma_1(\lceil (1-\varepsilon)N \rceil) = T\gamma_1\left( \left\lceil (1-\varepsilon)\frac{T}{K} \right\rceil \right) \leqslant cT \left( \left\lceil (1-\varepsilon)\frac{T}{K} \right\rceil \right)^{-\beta} \leqslant cT \left( (1-\varepsilon)\frac{T}{K} \right)^{-\beta}. \tag{49}$$

Thus, we enforce:

$$cT \left( (1-\varepsilon)\frac{T}{K} \right)^{-\beta} \leqslant \frac{\Delta_{(2)}(T)}{2} \implies T \geqslant 2^{\frac{1}{\beta-1}} \, c^{\frac{1}{\beta-1}} (1-\varepsilon)^{-\frac{\beta}{\beta-1}} \, K^{\frac{\beta}{\beta-1}} \, \Delta_{(2)}^{-\frac{1}{\beta-1}}(T).$$

$$\square$$

Now, we can find the error probability $e_T(\boldsymbol{\nu}, RR\text{-}SW)$, which will hold for all the time budgets which satisfy the condition of Lemma F.2.

**Theorem F.1.** *Under Assumptions 2.1 and 2.2, considering a time budget $T$ satisfying:*

$$T \geqslant 2^{\frac{1}{\beta-1}} \, c^{\frac{1}{\beta-1}} (1-\varepsilon)^{-\frac{\beta}{\beta-1}} \, K^{\frac{\beta}{\beta-1}} \, \Delta_{(2)}^{-\frac{1}{\beta-1}}(T), \tag{48}$$

*the error probability of* RR-SW *is bounded by:*

$$e_T(\boldsymbol{\nu}, RR\text{-}SW) \leqslant (K-1) \, \exp\left( -\frac{\varepsilon \, T}{8 \, K \, \sigma^2} \, \Delta_{(2)}^2(T) \right).$$

*Proof.* The RR-SW algorithm makes an error in predicting the best arm when, at the end of the process (at $T$ total pulls), the optimal arm has a pessimistic estimator $\hat{\mu}_1(N)$ that is not the highest among the arms (we consider w.l.o.g. that the best arm is the arm 1). Formally:

$$e_T(\boldsymbol{\nu}, RR\text{-}SW) = \underset{\boldsymbol{\nu}, \mathfrak{A}}{\mathbb{P}} (\exists i \in [\![2, K]\!] \; : \; \hat{\mu}_1(N) < \hat{\mu}_i(N))$$

$$\leqslant \sum_{i \in [\![2,K]\!]} \mathbb{P}_{\boldsymbol{\nu},\mathfrak{A}} \left( \hat{\mu}_1(N) < \hat{\mu}_i(N) \right).$$

Let us focus on a single arm $i \in [\![2, K]\!]$ and we want to upper bound the probability that $\mathbb{P}_{\boldsymbol{\nu},\mathfrak{A}} \left( \hat{\mu}_1(N) < \hat{\mu}_i(N) \right)$. Let us focus on the term inside the probability:

$$\hat{\mu}_i(N) \geqslant \hat{\mu}_1(N)$$

$$\hat{\mu}_i(N) - \hat{\mu}_1(N) \geqslant 0$$

$$\mu_1(T) - \hat{\mu}_1(N) + \hat{\mu}_i(N) - \mu_i(T) \geqslant \Delta_i(T) \tag{50}$$

$$\underbrace{\mu_1(T) - \bar{\mu}_1(N)}_{\leqslant T\gamma_1(\lceil (1-\varepsilon)N \rceil)} - \hat{\mu}_1(N) + \bar{\mu}_1(N) + \hat{\mu}_i(N) \underbrace{- \mu_i(T)}_{\leqslant -\bar{\mu}_i(N)} \geqslant \Delta_i(T) \tag{51}$$

$$-\hat{\mu}_1(N) + \bar{\mu}_1(N) + \hat{\mu}_i(N) - \bar{\mu}_i(N) \geqslant \Delta_i(T) - T\gamma_1(N-1), \tag{52}$$

where we added $\pm\Delta_i(T)$ to derive Equation 50, and added $\pm\bar{\mu}_1(N)$ to derive Equation 51, we used the results in Lemma E.8 and from the fact that the reward function is increasing. Considering a time budget $T$ satisfying Theorem F.2, and $\Delta_i(T) \geqslant \Delta_{(2)}(T)$ for all $i \in [\![2, K]\!]$, we have that:

$$T\gamma_1(\lceil (1-\varepsilon)N \rceil) \leqslant \frac{\Delta_i(T)}{2}. \tag{53}$$

Equation 53 holds since we are considering a time budget $T$ which satisfies a more restrictive condition (we are considering a time budget at which this separation already holds for $\lceil (1-\varepsilon)N \rceil$, so it also holds now). Substituting Equation 53 into Equation 52 the above, we have:

$$-\hat{\mu}_1(N) + \bar{\mu}_1(N) + \hat{\mu}_i(N) - \bar{\mu}_i(N) \geqslant \frac{\Delta_i(T)}{2},$$

and the error probability becomes:

$$e_T(\boldsymbol{\nu}, \text{RR-SW}) \leqslant \sum_{i=2}^{K} \mathbb{P}_{\boldsymbol{\nu},\mathfrak{A}} \left( -\hat{\mu}_1(N) + \bar{\mu}_1(N) + \hat{\mu}_i(N) - \bar{\mu}_i(N) \geqslant \frac{\Delta_i(T)}{2} \right).$$

For the previous argumentation, we apply the Azuma-Hoëffding's inequality and the union bound:

$$e_T(\boldsymbol{\nu}, \text{RR-SW}) \leqslant \sum_{i=2}^{K} \exp\left( -\frac{\varepsilon N \left( \frac{\Delta_i(T)}{2} \right)^2}{2\sigma^2} \right)$$

$$\leqslant (K-1) \exp\left( -\frac{\varepsilon T}{8K\sigma^2} \Delta_{(2)}^2(T) \right).$$

$\square$

## G   ON THE NECESSITY OF CONCAVITY ASSUMPTION

In Assumption 2.1, we introduce two conditions on the behavior of the expected payoffs over time: they must be non-decreasing and concave. If even one of these two lacks, the error probability cannot be guaranteed to be decreasing as a function of the budget $T$. From an intuitive perspective, this is similar to what happens for regret minimization in (Metelli et al., 2022, Theorem 4.2), in which the authors demonstrate the non-learnability (i.e., an $\Omega(T)$ cumulative regret lower bound) when these two assumptions do not hold.

From a technical perspective, we can demonstrate that the error probability no longer depends on $T$ when we remove the *concavity* assumption, as stated in the proposition below.

|        | $b$ | $c$  | $\psi$ |
|--------|-----|------|--------|
| Arm 1  | 37  | 1    | 1      |
| Arm 2  | 10  | 0.88 | 1      |
| Arm 3  | 1   | 0.78 | 1      |
| Arm 4  | 10  | 0.7  | 1      |
| Arm 5  | 20  | 0.5  | 1      |

Table 3: Numerical values of the parameters characterizing the functions for the synthetically generated setting.

**Proposition G.1.** *For every algorithm $\mathfrak{A}$, there exists a SRB $\boldsymbol{\nu}$ satisfying the non-decreasing assumption ($\gamma_i(n) \geqslant 0$, $\forall i \in [\![K]\!], n \in [\![T]\!]$), such that the error probability is lower bounded by:*

$$e_T(\boldsymbol{\nu}, \mathfrak{A}) \geqslant \frac{1}{4} \exp\left(-\frac{1}{8}\right).$$

*Proof.* Consider two Gaussian bandits with unit variance. Let $\boldsymbol{\nu}$ be a 2-armed bandit with expected rewards $\mu_1(n) = 1/2$ and $\mu_2(n) = 3/4$, both $\forall n \in [\![T]\!]$, thus its optimal arm is $i_{\boldsymbol{\nu}}^*(T) = 2$. Let $\boldsymbol{\nu}'$ be a 2-armed bandit with expected values $\mu_1(n) = 1/2 \ \forall n \in [\![T-1]\!]$, $\mu_1(T) = 1$ and $\mu_2(n) = 3/4 \ \forall n \in [\![T]\!]$, thus the optimal arm is $i_{\boldsymbol{\nu}'}^*(T) = 1$. Notice that bandit $\boldsymbol{\nu}'$ violates the concavity assumption. Now, applying the Bretagnolle-Huber inequality, we have that:

$$e_T(\boldsymbol{\nu}, \mathfrak{A}) = \max\left\{ \mathbb{P}_{\boldsymbol{\nu}, \mathfrak{A}}(\hat{I}^*(T) \neq 2), \mathbb{P}_{\boldsymbol{\nu}', \mathfrak{A}}(\hat{I}^*(T) \neq 1) \right\} \tag{54}$$

$$\geqslant \frac{1}{4} \exp\left(-\mathbb{E}_{\boldsymbol{\nu}, \mathfrak{A}}\left[\sum_{t=1}^{T} D_{\mathrm{KL}}(\nu_{I_t}(N_{I_t,t}), \nu'_{I_t}(N_{I_t,t}))\right]\right) \tag{55}$$

$$\geqslant \frac{1}{4} \exp\left(-D_{\mathrm{KL}}(\nu_1(T), \nu'_1(T))\right) \tag{56}$$

$$= \frac{1}{4} \exp\left(-\frac{1}{8}\right), \tag{57}$$

having observed that $D_{\mathrm{KL}}(\nu_{I_t}(N_{I_t,t}), \nu'_{I_t}(N_{I_t,t})) = 0$ if $t < T$ regardless the arm $I_t \in \{1, 2\}$ and that $D_{\mathrm{KL}}(\nu_{I_T}(N_{I_T,T}), \nu'_{I_T}(N_{I_T,T})) \leqslant D_{\mathrm{KL}}(\nu_1(T), \nu'_1(T)) = 1/8$. $\quad\square$

## H   EXPERIMENTAL DETAILS

In this section, we provide all the details about the presented experiments.

The payoff functions characterizing the arms shown in Figure 2 belong to the family:

$$F = \left\{ f(x) = c\left(1 - \frac{b}{(b^{1/\psi} + x)^{\psi}}\right) \right\},$$

where $c, \psi \in (0, 1]$ and $b \in [0, +\infty)$. Note that, by construction, all the functions laying in $F$ satisfy the Assumptions 2.1 and 2.2. In particular, the largest value of $\beta$ satisfying Assumption 2.2, for the setting presented in Section 8, is $\beta = 1.3$. In Table 3, we report the value of the parameters characterizing the function employed in the synthetically generated setting presented in the main paper.

### H.1   PARAMETERS VALUES FOR THE ALGORITHMS

This section provides a detailed view of the parameter values we employed in the presented experiments. More specifically, the parameters, which may still depend on the time budget $T$ and on the number of arms $K$, are set as follows:

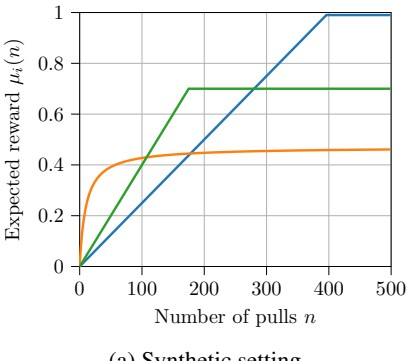

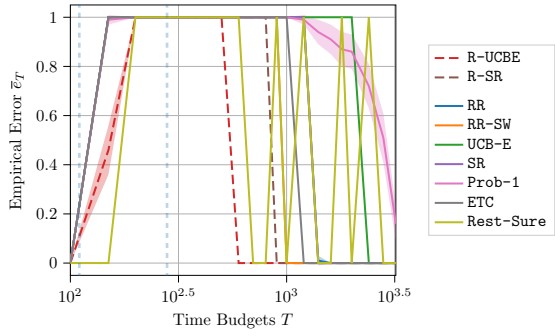

(a) Synthetic setting.    (b) Results on synthetic setting (100 runs, mean $\pm$ 95% c.i.).

Figure 6: Challenging scenario in which the arm reward increment rate changes abruptly.

- UCB-E: for the exploration parameter $a$, we used the optimal value, i.e., the one that minimizes the upper bound of the error probability, as prescribed in Audibert et al. (2010), formally:

$$a = \frac{25(T - K)}{36H_1},$$

where $H_1 = \sum_{i \neq i*(T)} \frac{1}{\Delta_i^2}$;

- R-UCBE: we used the value prescribed by Corollary 4.2 where we set the value $\beta = 1.3$;
- ETC and Rest-Sure: we set $\rho = 0.8$ and $U = 1$ as suggested by Cella et al. (2021).

### H.2 RUNNING TIME

The code used for the results provided in this section has been run on an Intel(R) I7 9750H @ 2.6GHz CPU with 16 GB of *LPDDR4* system memory. The operating system was *MacOS* 13.1, and the experiments were run on *Python* 3.10. A run of R-UCBE over a time budget of $T = 3200$ takes $\approx 0.07$ seconds (on average), while a run of R-SR takes $\approx 0.06$ seconds (on average).

## I ADDITIONAL EXPERIMENTAL RESULTS

In this section, we present additional results in terms of empirical error $\bar{e}_T$ of R-UCBE, R-SR, and the other baselines presented in Section 8.

### I.1 CHALLENGING SCENARIO

Here we test the algorithms on a challenging scenario in which we consider $K = 3$ arms whose increment changes *abruptly*. The setting is presented in Figure 6a. The results corresponding to such a setting are presented in Figure 6b. In this case, the last time the optimal arm does not change anymore is $T = 280$. Similarly to the synthetic setting presented in the main paper, we have two different behaviors for time budgets $T < 280$ and $T > 280$. For short time budgets, the algorithm providing the best performance is Rest-Sure, and the second best is R-UCBE. Conversely, for time budgets $T > 550$, R-UCBE provides a correct suggestion in most of the cases providing an error $\bar{e}_T(\boldsymbol{\nu}, \text{R-UCBE}) < 0.01$. Instead, Rest-Sure is not consistently providing reliable suggestions. This is allegedly due to the fact that such an algorithm has been designed to work in less general settings than the one we are tackling. Even in this case, the R-SR starts providing a small value for the error probability after R-UCBE does, at $T \approx 1000$. However, it is still better behaving than the other baseline algorithms. Note that Rest-Sure has a peculiar behavior. Indeed, it seems that even for large values of the time budget, it does not consistently suggest the optimal arm (i.e., the error probability does not go to zero). This is likely due to the nature of the parametric shape enforced by the algorithm, which may result in unpredictable behaviors when it does not reflect the nature of the real reward functions.

| | | | $T$ | | | | | | | | | | $R(\mathfrak{U})$ |
|---|---|---|---|---|---|---|---|---|---|---|---|---|---|
| | | 500 | 1000 | 2000 | 3000 | 4000 | 5000 | 7000 | 10000 | 15000 | 20000 | 30000 | |
| | Optimal Arm | 5 | 5 | 2 | 2 | 2 | 2 | 2 | 2 | 2 | 2 | 2 | |
| | R-UCBE (ours) | 6 | 5 | 2 | 5 | 2 | 2 | 2 | 2 | 2 | 2 | 2 | 9/11 |
| | R-SR (ours) | 5 | 5 | 5 | 5 | 5 | 5 | 5 | 5 | 2 | 2 | 2 | 5/11 |
| Algorithms | RR | 5 | 5 | 5 | 5 | 5 | 5 | 5 | 5 | 5 | 5 | 5 | 2/11 |
| | RR-SW | 5 | 5 | 5 | 5 | 5 | 5 | 5 | 5 | 5 | 2 | 2 | 4/11 |
| | SR | 5 | 5 | 5 | 5 | 5 | 5 | 5 | 5 | 5 | 5 | 2 | 3/11 |
| | UCB-E | 5 | 5 | 5 | 5 | 5 | 5 | 5 | 5 | 5 | 5 | 5 | 2/11 |
| | Prob-1 | 1 | 5 | 2 | 5 | 5 | 5 | 5 | 1 | 5 | 6 | 2 | 3/11 |
| | ETC | 5 | 5 | 5 | 5 | 5 | 5 | 5 | 5 | 5 | 5 | 2 | 3/11 |
| | Rest-Sure | 6 | 5 | 2 | 2 | 2 | 1 | 0 | 2 | 5 | 0 | 2 | 6/11 |

Table 4: Optimal arm for different time budgets on the IMDB dataset (first row) and corresponding recommendations provided by the algorithms (second to last row). In the last column, we compute the corresponding success rate.

## I.2 SENSITIVITY ANALYSIS ON THE NOISE VARIANCE

In what follows, we report the analysis of the robustness of the analyzed algorithms as noise standard deviation $\sigma$ changes in the collected samples. The setting we considered is the one described in Section 8. The results are provided in Figure 7. Let us focus on the performances of the R-UCBE algorithm. For small values of the standard deviation ($\sigma < 0.01$), we have the same behavior in terms of error probability, i.e., a progressive degradation of the performances for time budget $T = 150$. Indeed, at this time budget, the expected rewards of 3 arms are close to each other, and determining the optimal arm is a challenging problem. However, the performances are better or equal to all the other algorithms even at this point. Conversely, for values of the standard deviation $\sigma \geqslant 0.05$, the performance of R-UCBE starts to degrade, with behavior for $\sigma = 0.5$ which is constant w.r.t. the chosen time budget with a value of $\overline{e}_T(\boldsymbol{\nu}, \text{R-UCBE}) = 0.8$. This suggests that such an algorithm suffers in the case the stochasticity of the problem is significant. Let us focus on R-SR. This algorithm does not change its performances w.r.t. changes in terms of $\sigma$. Indeed, only for $\sigma = 0.5$, we have that it does not provide an error probability close to zero for time budget $T > 1000$. However, excluding R-UCBE, we have that the R-SR algorithm is the best/close to the best performing algorithm. This is also true in the case of $\sigma = 0.5$, in which the R-UCBE fails in providing a reliable recommendation for the optimal arm with a large probability.

## I.3 REAL-WORLD EXPERIMENT ON IMDB DATASET

**Description** We validate our algorithms and the baselines on an AutoML task, namely an *online best model selection* problem with a real-world dataset. We employ the IMDB dataset, made of $50,000$ reviews of movies (scores from 0 to 10). We preprocessed the data as done by Metelli et al. (2022), and run the algorithms for time budgets $T \in \mathcal{T} := \{500, 1000, \ldots, 15000, 20000, 30000\}$. A graphical representation of the reward (in this case, represented by the accuracy) of the different models is presented in Figure 8. Since, in this case, we only had a single realization to estimate the error probability $\overline{e}_T(\boldsymbol{\nu}, \mathfrak{A})$, we report the success rate $R(\boldsymbol{\nu}, \mathfrak{A})$ instead, i.e., the ratio between the number of times an algorithm provides a correct suggestion and the number of budget values we considered, formally defined as $R(\boldsymbol{\nu}, \mathfrak{A}) := \frac{1}{|\mathcal{T}|} \sum_{T \in \mathcal{T}} \mathbb{1}\{\hat{I}^*(T) = i^*(T)\}$ (the larger, the better).

**Results** The results are reported in Table 4. The algorithm with the largest success rate $R(\boldsymbol{\nu}, \mathfrak{A})$ is the R-UCBE, while R-SR provides the third best success rate. Moreover, Rest-Sure, the only algorithm providing a success rate larger than R-SR, has issues with large time budgets since for $T \geqslant 5000$ is able to provide only 2 correct guesses of the optimal arm over 6 attempts. Conversely, our algorithms progressively provide more and more correct guesses as the time budget $T$ increases. The above results on a real-world dataset corroborate the evidence presented above that the proposed algorithms outperform state-of-the-art ones for the BAI problem in SRB.

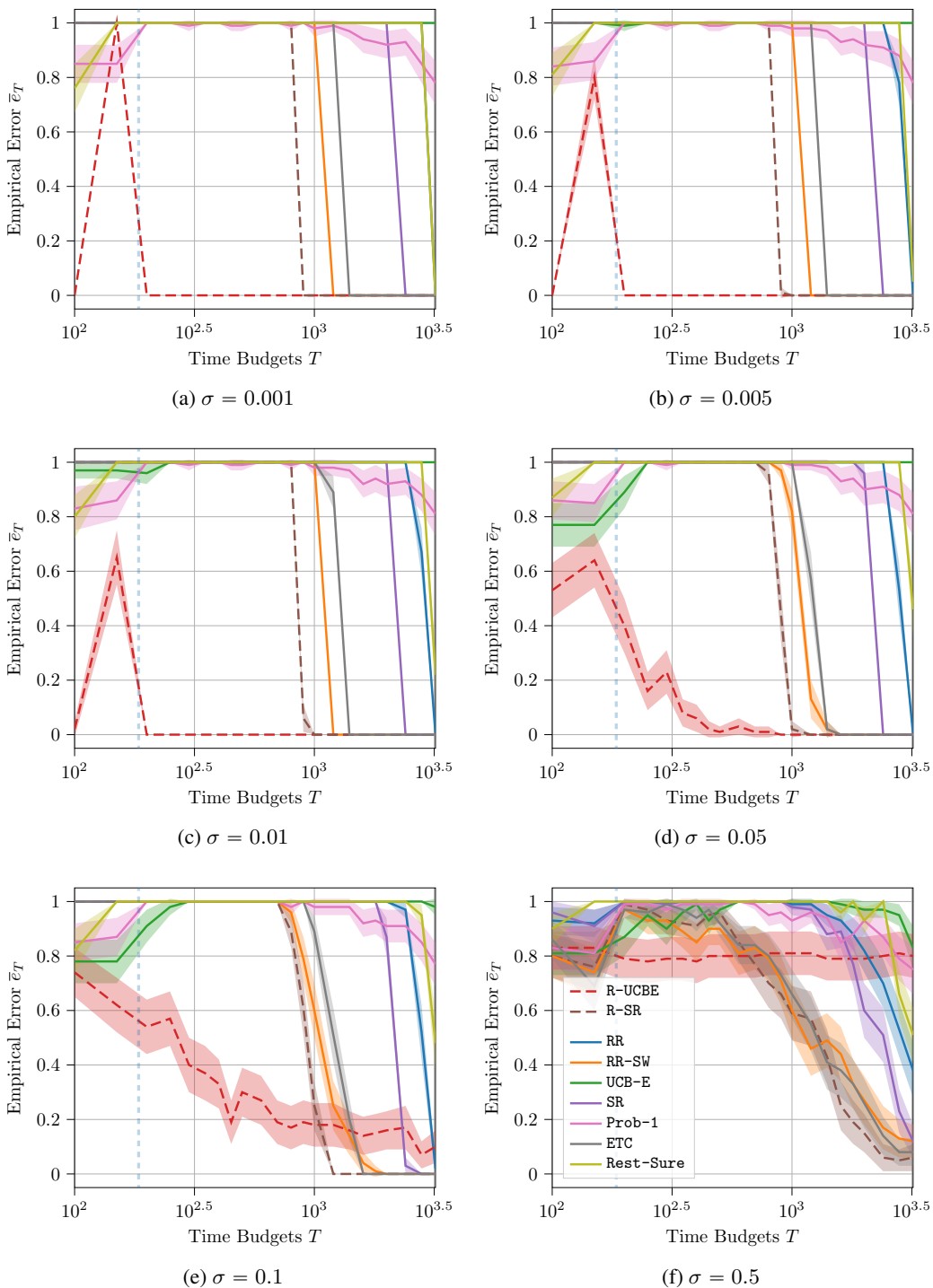

Figure 7: Empirical error probability for the synthetically generated setting, with different values of the noise standard deviation $\sigma$ (100 runs, mean $\pm$ 95% c.i.).

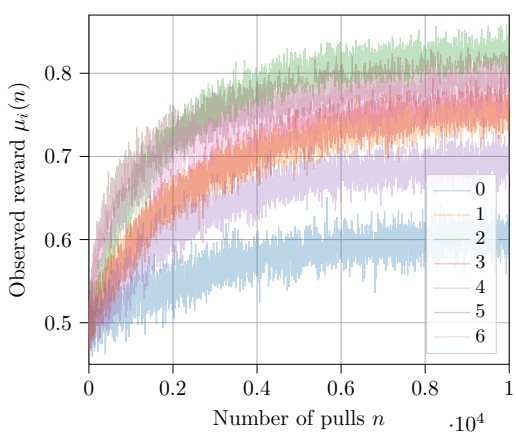

Figure 8: Rewards for the arms of the IMDB experiments.

