# OpenReview forum: "Best Arm Identification for Stochastic Rising Bandits"
_ICLR.cc/2024/Conference — Submitted to ICLR 2024_

### Official Review · Reviewer_nQcR · 2023-10-25

**Soundness:** 3 good
**Presentation:** 4 excellent
**Contribution:** 3 good
**Rating:** 5
**Confidence:** 4

**Summary:**

This paper considers the problem of fixed-budget best arm identification (BAI) within stochastic rising bandits. Specifically, it introduces both pessimistic and optimistic estimators for algorithm design. Building upon these estimators, two distinct algorithms emerge: R-UCBE and R-SR, drawing inspiration from UCB-E and SR as presented in Audibert et al. (2010). Regarding theoretical findings, the authors provide guarantees on the error probability of the two algorithms and investigate the minimal time budget $T$ required for the BAI task. Finally, numerical experiments conducted on synthetically generated data as well as a practical online best model selection problem serve to affirm the superiority of the proposed algorithms.

**Strengths:**

1. This paper is clear and well-organized.
2. The theoretical guarantee is exhaustive. Both the error probability and the minimum required time budget to accurately identify the optimal arm are taken into account.
3. The numerical experiments are impressive and comprehensive. The proposed algorithms clearly outperform the baselines.

**Weaknesses:**

1. While Assumption 2.1 appears intuitive, Assumption 2.2 falls short of being satisfactory. Even though the authors present some theoretical findings solely under Assumption 2.1, their interpretability is somewhat lacking.

2. The proposed algorithm closely resembles UCB-E and SR for standard multi-armed bandits, with the primary distinction lying in the estimators. I'm not suggesting this is unacceptable, but it does somewhat diminish the novelty of this work. A promising future direction would involve integrating both estimators into a unified algorithm.

3. Since the expected rewards are bounded in $[0,1]$ and non-decreasing, they must converge to some value. Thus, it is not surprising that the error probability lower bound will be matched by R-SR for large $T$. For non-stationary BAI, the algorithm SR is minimal optimal up to constant factors.

**Questions:**

1. Theorem 6.1: In any case, the algorithm can make a random guess. Therefore, it is not appropriate to state that $e_T(\boldsymbol{\nu}, \mathfrak{A})=1$.

2. Minor issue in Section 2: \citet should be used in "As in (Metelli et al., 2022)".

3. Figure 3: Could you elucidate some intuitions/explanations behind the remarkable performance of R-UCBE in cases where $T$ is small? The error probability approaches zero very quickly.

---

> ### Author Response · Authors · 2023-11-14
> **Response to Reviewer nQcR (1/2)**
>
> We thank the Reviewer for spending time revising our work, for having appreciated our theoretical analysis and our experimental campaign. Below, our response to the reviewer's concerns.
>
> ### Weaknesses
>
> > While Assumption 2.1 appears intuitive, Assumption 2.2 falls short of being satisfactory. Even though the authors present some theoretical findings solely under Assumption 2.1, their interpretability is somewhat lacking.
>
> Assumption 2.1 implies that the expected values are *non-dereasing and concave*. Assumption 2.2 provides a more **explicit characterizarion** of the increment function $\gamma_i(n)$ by enforcing an upper bound of the form $c n^{-\beta}$ depending on the parameters $(c,\beta)$. Although our results hold under Assumption 2.1 only (Theorems 4.1 and 5.1), the forms of the error probability $e_T$ and of the minimum time budget $T$ become more interpretable under Assumption 2.2. Indeed, intuitively, **the BAI problem is easier when the expected reward function reaches the stationary behavior faster** (see [[IMAGE](https://drive.google.com/file/d/1vcthAAJtBTKybq1GULnNdUTk5EAo9kvj/view?usp=sharing)]). This "speed" (a.k.a. rate) is governed mainly by $\beta$. The larger is $\beta$, the faster is the convergence rate. This is visible in Corollary 4.2 and 5.2 where the minimum requested time budget $T$ as well as the error probability decrease as $\beta$ increase. We will add this comment in the final version.
>
> > The proposed algorithm closely resembles UCB-E and SR for standard multi-armed bandits, with the primary distinction lying in the estimators. I'm not suggesting this is unacceptable, but it does somewhat diminish the novelty of this work. <!--A promising future direction would involve integrating both estimators into a unified algorithm.-->
>
> While we agree that our algorithms follow the basic scheme of the ones of (Audibert et al., 2010), we point out that our algorithms follow **well-established principles**: *optimism in the face of uncertainty* (UCB-E) and *arm elimination* (SR) which are the basic building blocks of a large majority of works in bandits published in top-tier venues [1,2,3]. Thus, we do not think this diminishes the novelty of the work. Furthermore, we stress that our main contributions are: ($i$) the challenging **theoretical analysis** that the rested rising nature of the arms requires for both the estimators and the algorithms and ($ii$) the conception of the novel **lower bound** which shed lights on the complexities of the setting.
>
> > Since the expected rewards are bounded in $[0,1]$ and non-decreasing, they must converge to some value. Thus, it is not surprising that the error probability lower bound will be matched by R-SR for large $T$. For non-stationary BAI, the algorithm SR is minimal optimal up to constant factors.
>
> While we agree that the expected rewards must converge to some value, it is not guaranteed that they will all converge to **different values**. Suppose that the second-optimal arm converges for $T \rightarrow +\infty$ to the same expected reward as the optimal arm. This SRB instance will be very challenging for large values of $T$. This is why we need a characterization of the conditions (Eq. 11 or 13) under which R-SR succeeds in identifying the optimal arm. Furthermore, we remark that the application of standard SR is not guaranteed to yield the same guarantees, since it uses as estimator the **standard sample mean** that does not discard too old samples and might fail to deliver an estimator with a bias that diminishes at the number of samples increases.

---

> > ### Author Response · Authors · 2023-11-14
> > **Response to Reviewer nQcR (2/2)**
> >
> > ### Questions
> >
> > > Theorem 6.1: In any case, the algorithm can make a random guess. Therefore, it is not appropriate to state that $e_T(\mathbf{\nu}, \mathfrak{A}) = 1$.
> >
> > We agree with the reviewer that such an expression is not appropriate. We will remove it in the final version.
> >
> > > Minor issue in Section 2: \citet should be used in "As in (Metelli et al., 2022)".
> >
> > Thank you, we fixed it.
> >
> > > Figure 3: Could you elucidate some intuitions/explanations behind the remarkable performance of R-UCBE in cases where $T$ is small? The error probability approaches zero very quickly.
> >
> > First of all, as also for Figure 4, the error probability increases up to $T=185$ and then falls quickly to zero. This is due to the change of the optimal arm, indicated by the vertical dashed blue line. By looking at Figure 2, we can observe how for $T < 185$ the best arm is the orange one, while for $T \geq 185$ it becomes the blue one. As the time horizon approaches such a value of $T$, the algorithm predicts the blue arm as the optimal one, even if the real optimal is the orange, for this reason the error probability increases. Then, for $T \geq 185$ the real optimal arm is the blue one and the algorithm predicts, as done as much $T$ approaches $185$ from the left. Moreover, the R-UCBE algorithm seems to be the one performing better, i.e., it is the one with the error rate that approaches zero in the fastest way. This happens in the case presented in the main paper. However, in the Appendix I, we can observe how R-UCBE performs worst when $\sigma$ increases compared to R-SR (Figure 7 (e) and (f)). Thus, we can interpret this behavior of R-UCBE as the effect a possible *underexploration* that determines very good results when the noise magnitude is small, but fails to deliver the best arm when the reward is affected by high noise. We will remark this in the final version.
> >
> > ---
> >
> > [1] Gabillon, V., Ghavamzadeh, M., Lazaric, A., & Bubeck, S. (2011). Multi-bandit best arm identification. Advances in Neural Information Processing Systems, 24.
> >
> > [2] Jamieson, K., & Talwalkar, A. (2016). Non-stochastic best arm identification and hyperparameter optimization. In Artificial intelligence and statistics (pp. 240-248). PMLR.
> >
> > [3] Gao, Z., Han, Y., Ren, Z., & Zhou, Z. (2019). Batched multi-armed bandits problem. Advances in Neural Information Processing Systems, 32.

---

> ### Comment · Reviewer_nQcR · 2023-11-15
>
> I appreciate your prompt response.
>
> - I still hold the opinion that Assumption 2.2 is excessively strong and unrealistic, despite the proposed algorithms not requiring it. The absence of Assumption 2.2 renders the theoretical results perplexing and challenging to comprehend. Are there any analogous assumptions in existing literature that might provide a basis for comparison?
>
> - Upon reading all other reviews and their corresponding rebuttals, I find myself somewhat perplexed by the problem setup. Why is there a need to explore BAI in stochastic rising bandits? The definition of the best arm in this context appears intricate, and even if we find the best arm, pulling it is essential for improving its expected reward, which strikes me as unusual. What advantages does BAI offer over regret minimization in the context of stochastic rising bandits?

---

> > ### Author Response · Authors · 2023-11-16
> > **Response to Reviewer nQcR**
> >
> > We thank the Reviewer for the prompt reply.
> >
> > > I still hold the opinion that Assumption 2.2 is excessively strong and unrealistic, despite the proposed algorithms not requiring it. The absence of Assumption 2.2 renders the theoretical results perplexing and challenging to comprehend. Are there any analogous assumptions in existing literature that might provide a basis for comparison?
> >
> > We disagree on the fact that Assumption 2.2 is "excessively strong and unrealistic". Indeed, there are examples of analogous assumptions in the existing literature:
> > - The seminal paper [1] (*published at ICML 2022*) that studies regret minimization for stochastic rising bandits provides regret bounds that contain the $\Upsilon(T,q)$ term (see Theorem 4.4 and 5.3) that can be considered "challenging to comprehend" in the same way as our results under Assumption 2.1 only. Indeed, similarly to our Assumption 2.2, in Table 1, the authors of [1] provide a bound to the $\Upsilon(T,q)$ term  **under the assumption that the increments satisfy $\gamma_i(l) \le l^{-c}$**. This corresponds to our Assumption 2.2 with $n \leftarrow l$, $\beta \leftarrow c$ and $c \leftarrow 1$ (in [1] also an increment bounded by $e^{-cl}$ is considered that is even stronger than $l^{-c}$). Thus [1] follows the very same rationale of providing more interpretable results under an assumption analogous to Assumption 2.2.
> > - The paper [2] (*published at ICML 2021*), that studies best-arm identification in stochastic rising bandits, **assumes that the expected reward functions $\mu_i(n)$ follow a KNOWN functional form $\frac{\alpha_i}{(1+n)^\rho} + \beta_i$ with SOME UNKNOWN parameters, namely $\alpha$ and $\beta$, while $\rho$ is known**. As we already pointed out in Section 7, this assumption is **stronger than our Assumption 2.2**, since ours requires **no specific known functional form**.
> >
> > Furthermore, we remark that given that the condition of Assumption 2.2 is evaluated at discrete time steps, it is **always possible** to find a couple ($c,\beta$) for which every curve satisfies such an assumption.
> >
> > > Upon reading all other reviews and their corresponding rebuttals, I find myself somewhat perplexed by the problem setup. Why is there a need to explore BAI in stochastic rising bandits? The definition of the best arm in this context appears intricate, and even if we find the best arm, pulling it is essential for improving its expected reward, which strikes me as unusual. What advantages does BAI offer over regret minimization in the context of SRB?
> >
> > We think we have widely explained why the BAI problem in SRB is highly interesting in the paper. We reiterate these explanations here for the Reviewer's convenience:
> > - **CASH (or Best Model Selection)** (see also Section 1): In these real-world problems emerging from the AutoML applications, we are required to find the *best algorithms together with the best hyperparameter configuration*. Thus, each tuple (algorithm, hyperparameter configuration) corresponds to an arm whose performance increases on average whenever it is pulled, i.e., a unit of computation/batch of data is provided by the algorithm. In the CASH problem, we are interested in finding the one which **attains the best performance at the end of the budget $T$**. This justifies the **definition of optimal arm $i^*(T)$ we provide that, in light of this, is the natural choice for such a problem**.
> > - **BAI vs Regret minimization**: Furthermore, in problems like CASH, we are interested in minimizing the probability of recommending the wrong arm at the end of the budget. This corresponds to **minimize the amount of computation/data spent on suboptimal options** since, once the best option is identified, in a real-world scenario, it will be deployed. In this context, it is clear that we **do not care about the loss (regret) we suffer during the learning process**, as long as the optimal option is identified. This clearly explains why we are interested in BAI and not in regret minimization for such applications.
> > - To conclude, paper [2] (*published at ICML 2021*) considers the BAI problem for a **class of stochastic rising bandits** as well. **The fact that this paper is published at ICML 2021 justifies that studying the BAI problem for SRBs is interesting for the community.** We remark that [2] differs from our work for at least two reasons. First, [2] focuses on a subclass of stochastic rising bandits, more restrictive w.r.t. our Assumption 2.2. Second, it focuses on controlling the *simple regret*, while we provide bounds on the *error probability*, from which tight simple regret bounds follow (see Section 7 and Theorem E.10).
> >
> > We will make sure to further remark these aspects in the final version of the paper.
> >
> > [1] Metelli et al. "Stochastic rising bandits." International Conference on Machine Learning. PMLR, 2022.
> >
> > [2] Cella et al. "Best model identification: A rested bandit formulation." International Conference on Machine Learning. PMLR, 2021.

---

> > > ### Comment · Reviewer_nQcR · 2023-11-16
> > >
> > > Thank you for your comprehensive response. It's great to see that similar assumptions to Assumption 2.2 are already used in existing literature. I suggest the authors expand on the rationale behind Assumption 2.2 in the main text, akin to the approach taken for Assumption 2.1.
> > >
> > > Let's direct our attention to the weakness in the setting, which, in my opinion, is more significant. It seems there might be a misunderstanding of my previous point. In fact, my comment is closely related to that of Reviewer CfKv.
> > >
> > > The definition of the 'best arm' in this paper, referred to as arm 1, is based on $\mu_1(T)$, the expected reward from pulling arm 1 for $T$ times. Why is it meaningful to find such best arm in the fixed-budget setting?  In the CASH example, the authors state, “In the CASH problem, we are interested in finding the one which attains the best performance at the end of the budget T” and that “once the best option is identified, in a real-world scenario, it will be deployed”. However, successfully identifying the best arm does not equate to attaining the performance of $\mu_1(T)$. What we obtain is $\mu_1(\tau)$, where $\tau$ represents the number of times arm 1 is pulled up to time $T$. To realize $\mu_1(T)$, we need to run the tuple (algorithm, hyperparameter configuration) for $T-\tau$ additional rounds. Hence, this example does not justify the appropriateness of the problem setup.
> > >
> > > Regarding the reference paper [2], it actually supports my viewpoint. In the context of stochastic rising bandits, the simple regret minimization problem is natural. Translated into the notation used in this paper, the simple regret is defined as $\mu_1(T)-\mu_i(\tau_i)$, where $i$ is the algorithm's output, and $\tau_i$ is its number of arm pulls up to time $T$. This is evidently more practical.
> > >
> > > ---
> > >
> > > I am delighted to engage in further discussions with the authors regarding this issue.

---

> > > > ### Author Response · Authors · 2023-11-17
> > > > **Response to Reviewer nQcR**
> > > >
> > > > We thank the reviewer for this interaction that allow us to improve the presentation of our work.
> > > >
> > > > > I suggest the authors expand on the rationale behind Assumption 2.2 in the main text, akin to the approach taken for Assumption 2.1.
> > > >
> > > > We will integrate a detailed comment in the final version of the work.
> > > >
> > > > > It seems there might be a misunderstanding of my previous point. In fact, my comment is closely related to that of Reviewer CfKv. The definition of the 'best arm' in this paper, referred to as arm 1, is based on $\mu_1(T)$, the expected reward from pulling arm $1$ for $T$ times. Why is it meaningful to find such best arm in the fixed-budget setting? In the CASH example, the authors state, “In the CASH problem, we are interested in finding the one which attains the best performance at the end of the budget T” and that “once the best option is identified, in a real-world scenario, it will be deployed”. However, successfully identifying the best arm does not equate to attaining the performance of $\mu_1(T)$. What we obtain is $\mu_1(\tau)$, where $\tau$ represents the number of times arm 1 is pulled up to time $T$. To realize $\mu_1(T)$, we need to run the tuple (algorithm, hyperparameter configuration) for $T-\tau$ additional rounds. Hence, this example does not justify the appropriateness of the problem setup. Regarding the reference paper (Cella et al, 2021), it actually supports my viewpoint. In the context of stochastic rising bandits, the simple regret minimization problem is natural. Translated into the notation used in this paper, the simple regret is defined as $\mu_1(T) - \mu_i(\tau_i)$, where $i$ is the algorithm's output, and $\tau_i$ is its number of arm pulls up to time $T$. This is evidently more practical.
> > > >
> > > > Thank you for the answer, we have now gained a clearer understanding of the Reviewer's key point. Let us remark two aspects:
> > > > 1. **[Idenfifying $i^*(T)$ is meaningful and useful]** We agree with the reviewer that simple regret $\mu_{i^*(T)}(T) - \mu_i(\tau_i)$ is an appropriate performance index in the motivating examples provided. Focusing on the *online model selection* problem, an algorithm that controls the simple regret allows to answer the question **How much suboptimal is the performance $\mu_i(\tau_i)$ of the recommended model at the end of the training budget $T$ compared to the performance $\mu_{i^*(T)}(T)$ of the best model $i^*(T)$ trained from the beginning with the full training budget $T$?** This is indubitably a very important question and we remark that **we also answer to this question since our R-SR allows to control optimally the simple regret** (see point 2). Nevertheless, answering this question does not solve the problem: **Which is the model that would have best performed if trained from the beginning with the full training budget $T$?**. This question is significant as well. Indeed, remaining on the *online model selection* metaphor, suppose that we need to select a *model to be trained in multiple similar environments* with a budget $T$. We could set up an experiment to identify which model to be selected using one of our algorithms in one of the environments or in simulation and, then, once identified, perform multiple independent training in the real environments. In such a scenario, idenfifying $i^*(T)$ is meaningful and useful, as well as controlling the error probability. We will remark these considerations in the final version.
> > > > 5. **[R-SR Algorithms enjoys optimal simple regret bound]**  We also studied the performances of our top-performing solution (R-SR) in terms of *simple regret* precisely with the goal of making a comparison with the algorithm of (Cella et al, 2021). This discussion is provided at the end of Section 7. In this section, we refer to our **Theorem E.10, in which we show that we are matching the lower bound on the simple regret of (Cella et al, 2021), up to logarithmic factors**.
> > > >
> > > > ---
> > > >
> > > > Audibert, J. Y., Bubeck, S., & Munos, R. (2010). Best arm identification in multi-armed bandits. In Conference on Learning Theory (pp. 41-53).
> > > >
> > > > Carpentier, A., & Locatelli, A. (2016). Tight (lower) bounds for the fixed budget best arm identification bandit problem. In Conference on Learning Theory (pp. 590-604). PMLR.
> > > >
> > > > Cella, L., Pontil, M., & Gentile, C. (2021). Best model identification: A rested bandit formulation. In International Conference on Machine Learning (pp. 1362-1372). PMLR.

---

> > > > > ### Comment · Reviewer_nQcR · 2023-11-17
> > > > >
> > > > > Thank you for your response. I concur that the central focus of this paper is to address the question: which is the model that would have best performed if trained from the beginning with the full training budget T? Consequently, the illustrative examples provided in the paper may require modification for a more accurate representation. Additionally, I acknowledge that the problem studied is meaningful when there are multiple similar training environments, but this seems somewhat limiting.
> > > > >
> > > > > Based on the preceding discussions, I prefer to retain my current rating, although it could potentially change after the reviewer discussion phase.

---

### Official Review · Reviewer_BjA1 · 2023-10-27

**Soundness:** 4 excellent
**Presentation:** 2 fair
**Contribution:** 2 fair
**Rating:** 5
**Confidence:** 3

**Summary:**

This paper focus on the stochastic rising bandits, the objective is maximize the success rate of identifying the best arm within fixed budget. The authors propose two algorithms, and one of them is optimal in success rate as the authors further give the lower bound that matches the upper bound. Authors further conduct synthetic experiments to validate the theoretical findings.

**Strengths:**

- The theoretical proofs are strict and easy to follow, the results seem sound to me.
- The experiments are explicitly introduced with specific details.
- The guarantee of R-SR is strong, and the analysis on the minimum budget the problem is solvable is crucial to the problem, making it clear on which parts of the problem is unsolvable.

**Weaknesses:**

- My major concern is the insufficient problem motivation. In the introduction, the example introduces the SRB is ``the arm improve performances over time'', but the problem setup of SRB is arms whose performances increase with pulls. I personally feel it is the example of adversarial MAB or non-stationary MAB rather than SRB. The experiments still do not give the real-world applications. In fact it's hard for me to figure out real-world scenario (with the neccessary to model as a SRB) that solves practical problems.

- The problem statement is a little bit unclear. Specifically, it is mentioned that SRB is a special case of SRB, but it is never explained what the word ``rested'' means.

- There should be some discussions about the difficulties of applying existing algorithms (or some trivial variants) to solve SRB. For example, it is only mentioned in the experiments that non-stationary MAB algorithms and adversarial MAB algorithms is outperformed, but it is essential to verify that the increasing structure of SRB is crucial both theoretically and empirically.

**Questions:**

See above

---

> ### Author Response · Authors · 2023-11-14
> **Response to Reviewer BjA1**
>
> We thank the Reviewer for spending time revising our work, and for having appreciated our theoretical analysis and our experimental campaign. Below, our response to the reviewer's concerns.
>
> > My major concern is the insufficient problem motivation. In the introduction, the example introduces the SRB is "the arm improve performances over time", but the problem setup of SRB is arms whose performances increase with pulls. I personally feel it is the example of adversarial MAB or non-stationary MAB rather than SRB. In fact it's hard for me to figure out real-world scenario (with the neccessary to model as a SRB) that solves practical problems.
>
> In the main paper, we proposed a relevant *motivating practical example*, the **combined algorithm selection and hyperparameter optimization (CASH)**, **a real-world scenario** arising from the AutoML research field that **naturally models as a SRB problem** (see Introduction). Moreover, we devoted Appendix C to discuss **two additional** motivating examples. We refer the Reviewer to these sections (Introduction and Appendix C) for a more detailed discussion.
> We agree that the sentence "the arm improve performances over time" is misleading since it may suggest that the arms evolve *regardless* they are pulled. We will replace it in the final version of the paper with "the arms improve their performances as it is pulled" to avoid confusion.
>
> > The experiments still do not give the real-world applications. In fact it's hard for me to figure out real-world scenario (with the neccessary to model as a SRB) that solves practical problems.
>
> In the experimental campaign, we test our methods with **real-world data** by them to perform a **Best Model Selection** of a set of different machine learning algorithms trained to address the classification task on the IMDB dataset. Such an experiment is reported in Appendix I.3 due to space constraints.
>
> > The problem statement is a little bit unclear. Specifically, it is mentioned that SRB is a special case of SRB, but it is never explained what the word "rested" means.
>
> We think the Reviewer with the sentence  "it is mentioned that SRB is a special case of SRB" means "it is mentioned that SRB is a special case of **rested bandit**". We remark that **Section 1 (Introduction) does explain the word "rested"** precisely at the beginning of the second paragraph, which we report here for the Reviewer's convenience: "This work focuses on the Stochastic Rising Bandits (SRB), a specific instance of the rested bandit setting (Tekin & Liu, 2012) in which the expected reward of an arm increases whenever it is pulled".
>
>
> > There should be some discussions about the difficulties of applying existing algorithms (or some trivial variants) to solve SRB. For example, it is only mentioned in the experiments that non-stationary MAB algorithms and adversarial MAB algorithms is outperformed, but it is essential to verify that the increasing structure of SRB is crucial both theoretically and empirically.
>
> We considered several baselines in our numerical validation, including *UCB-E* (Audibert et al., 2010) and *Successive Rejects* (Audibert et al., 2010), the most famous solutions for fixed-budget BAI in stationary MABs. These solutions are based on **estimators that do not discard old data**, as they are developed for stationary setting, **preventing from obtaining estimator of $\mu_i(T)$ whose bias decreases** as the number of pulls increases. This prevents the error probability to decrease with $T$. We considered also *Prob-1* (Abbasi-Yadkori et al., 2018) as a baseline from the adversarial setting. However, we point out that in our case the rewards *stochastic and subgaussian* and not *bounded and adversarial*. Since *Prob-1* assumes bounded rewards, its analysis does not apply to our setting.
>
> ---
>
> Audibert, J. Y., Bubeck, S., & Munos, R. (2010). Best arm identification in multi-armed bandits. In COLT (pp. 41-53).
>
> Abbasi-Yadkori, Y., Bartlett, P., Gabillon, V., Malek, A., & Valko, M. (2018). Best of both worlds: Stochastic & adversarial best-arm identification. In COLT (pp. 918-949).

---

### Official Review · Reviewer_CfKv · 2023-10-30

**Soundness:** 3 good
**Presentation:** 3 good
**Contribution:** 3 good
**Rating:** 6
**Confidence:** 3

**Summary:**

This paper addresses the problem of best arm identification in the context of stochastic rising bandits with a fixed budget, aiming to identify the arm with the maximum expected reward in the final round. Two algorithms are proposed to tackle this issue: one is a UCB-typed algorithm, and the other is a successive-reject-typed algorithm. The paper also establishes a sample number lower bound for BAI problem of SRB setting, as well as an error lower bound when the sample number is fixed. The theoretical guarantees obtained show that R-UCBE is optimal but requires additional prior knowledge, while R-SR reduces the dependence on prior knowledge. Empirical results further demonstrate that R-UCBE and R-SR outperform other algorithms in comparison.

**Strengths:**

This work is clearly written and provides two solid approaches supported by theory and experiments. It also offers lower bounds for the problem, making it a fairly complete piece of work.

**Weaknesses:**

Assuming there is a unique best arm seems somewhat unrealistic, especially after T rounds, when there is a high probability that multiple arms could have the same reward. This can be observed in Figure 2 of the experiment, where several lines easily overlap, clearly demonstrating this point. Moreover, this situation is influenced by the randomness of the algorithm, similar to the paper's mention that "$i^*(T)$ may change," which is also a result of the algorithm's randomness. While similar assumptions are made in classical MAB settings, in those cases, the algorithm does not have an impact on the best arm.

**Questions:**

see the weakness

---

> ### Author Response · Authors · 2023-11-14
> **Response to Reviewer CfKv**
>
> We thank the Reviewer for spending time reviewing our work, and for having appreciated our clarity and completeness. Below, our answers to the reviewer's concerns.
>
> > Assuming there is a unique best arm seems somewhat unrealistic, especially after T rounds, when there is a high probability that multiple arms could have the same reward. This can be observed in Figure 2 of the experiment, where several lines easily overlap, clearly demonstrating this point.
>
> The assumption that the optimal arm is unique is **standard** in the bandit best arm identification community. Such an assumption is present in the seminal paper in this setting (Audibert et al., 2010) and in (Lattimore and Szepesvári, 2020, Chapter 33). Moreover, in our setting, it is proved to be **necessary** to make the problem learnable.
> We fear that the Reviewer might have misunderstood what we mean for "unique optimal arm". Indeed, we have to enforce that **at the end of the budget $T$** the arm with the hightest expected reward (i.e., the optimal) is unique and **not that there are no intersection over the full horizon $T$** (and so we admit the same expected reward at some points). Looking at Figure 2, for $T \in$ {$1,\dots,300$}, we notice that there are 4 points of intersection. Thus, there are just **4 choices of $T$ over 300** in which the optimal arm is not unique. We believe that this is a light standard assumption.
>
> > Moreover, this situation is influenced by the randomness of the algorithm, similar to the paper's mention that "$i^*(T)$ may change", which is also a result of the algorithm's randomness. While similar assumptions are made in classical MAB settings, in those cases, the algorithm does not have an impact on the best arm.
>
> Given a specific time budget $T$ (which is an input of the fixed-budget BAI problem) and given an instance of the stochastic rising bandit (i.e., the expected rewards $\mu_i(t)$ functions), the optimal arm $i^*(T) = \text{argmax}_{i \in [K]} \mu_i(T)$ is well-defined and, thus, **algorithm-independent** (being defined through $T$ and $\mu_i(t)$ irrespective of the used algorithm). We stress that, **exactly like in classical MAB settings, the algorithm does not have an impact on the best arm also in stochastic rising bandits**.
>
> ---
>
> Audibert, J. Y., Bubeck, S., & Munos, R. (2010). Best arm identification in multi-armed bandits. In COLT (pp. 41-53).
>
> Lattimore, T., & Szepesvári, C. (2020). Bandit algorithms. Cambridge University Press.

---

### Meta-Review · Area_Chair_BUMc · 2023-12-05

**Metareview:**

This is a borderline paper. The reviewers have, in particular, concerns about the problem setting; these are about clarity of definitions and description as well as the usefulness of the setting. I feel that these need to be addressed properly before the paper is ready for publication.

**Justification For Why Not Higher Score:**

This is a borderline paper and I was between a weak accept and a weak reject.

**Justification For Why Not Lower Score:**

n/a

---

### Decision · Program_Chairs · 2024-01-16

Reject